# PNA6, a Lactosyl Analogue of Angiotensin-(1-7), Reverses Pain Induced in Murine Models of Inflammation, Chemotherapy-Induced Peripheral Neuropathy, and Metastatic Bone Disease

**DOI:** 10.3390/ijms241915007

**Published:** 2023-10-09

**Authors:** Maha I. Sulaiman, Wafaa Alabsi, Lajos Szabo, Meredith Hay, Robin Polt, Tally M. Largent-Milnes, Todd W. Vanderah

**Affiliations:** 1Department of Pharmacology, College of Medicine, The University of Arizona, Tucson, AZ 85721, USA; mahasulaiman81@arizona.edu (M.I.S.); tlargent@arizona.edu (T.M.L.-M.); 2Department of Chemistry & Biochemistry, The University of Arizona, Tucson, AZ 85721, USA; wafaaalabsi@arizona.edu (W.A.); lzszabo@arizona.edu (L.S.); polt@arizona.edu (R.P.); 3Skaggs Pharmaceutical Sciences Center, College of Pharmacy, The University of Arizona, 1703 E. Mabel St, Tucson, AZ 85721, USA; 4The BIO5 Institute, The University of Arizona, Tucson, AZ 85721, USA; mhay@arizona.edu; 5Department of Physiology, The University of Arizona, Tucson, AZ 85721, USA; 6Evelyn F. McKnight Brain Institute, The University of Arizona, Tucson, AZ 85721, USA; 7Comprehensive Pain and Addiction Center, University of Arizona, Tucson, AZ 85721, USA

**Keywords:** PNA6, Angiotensin-(1-7), mas receptor, breast, cancer, pain

## Abstract

Pain is the most significant impairment and debilitating challenge for patients with bone metastasis. Therefore, the primary objective of current therapy is to mitigate and prevent the persistence of pain. Thus, cancer-induced bone pain is described as a multifaceted form of discomfort encompassing both inflammatory and neuropathic elements. We have developed a novel non-addictive pain therapeutic, PNA6, that is a derivative of the peptide Angiotensin-(1-7) and binds the Mas receptor to decrease inflammation-related cancer pain. In the present study, we provide evidence that PNA6 attenuates inflammatory, chemotherapy-induced peripheral neuropathy (CIPN) and cancer pain confined to the long bones, exhibiting longer-lasting efficacious therapeutic effects. PNA6, Asp-Arg-Val-Tyr-Ile-His-Ser-(O-β-Lact)-amide, was successfully synthesized using solid phase peptide synthesis (SPPS). PNA6 significantly reversed inflammatory pain induced by 2% carrageenan in mice. A second murine model of platinum drug-induced painful peripheral neuropathy was established using oxaliplatin. Mice in the oxaliplatin-vehicle treatment groups demonstrated significant mechanical allodynia compared to the oxaliplatin-PNA6 treatment group mice. In a third study modeling a complex pain state, E0771 breast adenocarcinoma cells were implanted into the femur of female C57BL/6J wild-type mice to induce cancer-induced bone pain (CIBP). Both acute and chronic dosing of PNA6 significantly reduced the spontaneous pain behaviors associated with CIBP. These data suggest that PNA6 is a viable lead candidate for treating chronic inflammatory and complex neuropathic pain.

## 1. Introduction

Breast cancers are the most frequent malignant tumors in women of all racial and ethnic backgrounds in the United States and the second leading cause of death in women [1]. More than 1.3 million women are diagnosed with breast cancer each year. In 2019, the Centers for Disease Control (CDC) reported 264,121 new cases of female breast cancer, with 42,280 deaths in total (CDC, 2019). Breast cancer can also occur in men, although men represent 1% of all breast cancer cases [2]. 

Of the patients who die from breast cancer, greater than 90% have been associated with the progression of metastatic breast cancer [3]. The brain, lungs, liver, and bone are the most common loci for breast tumor metastasis. Clinically, bone metastasis represents 60–70% of all breast cancer metastasis [4,5]. Over 50% of individuals with advanced breast cancer have bone metastasis, and about 70% of women who die from breast cancer develop bone metastases [6,7,8]. Typical complications of bone metastasis are pathological fracture, spinal compression, hypercalcemia, and persistent pain [9]. These complications lead to a massive reduction in the survival rate and a decreased quality of life.

Pain is the most significant impairment and debilitating challenge for patients with bone metastasis. Therefore, the primary objective of current therapy is to mitigate and prevent the persistence of pain. Cancer-induced bone pain can be intermittent but progresses rapidly into continuous pain exacerbated by episodes of breakthrough pain. Thus, cancer-induced bone pain is characterized as a complex pain comprising inflammatory, neuropathic, and mechanical components, which refers to pain that is associated with physical changes or mechanical factors within the body, such as the compression of nerves or tissues by tumors or the expansion of bone due to metastatic growth [5,10,11]. In addition to cancer-induced pain, chemotherapy often results in long-lasting neuropathic pain that can last well after treatment [12]. Chemotherapy-induced peripheral neuropathy (CIPN) is a progressive, durable, and often irreversible side effect of various antineoplastic medications. Thus, interventions that can alleviate both cancer-induced and chemotherapy-related neuropathic pain are highly desirable.

One class of antineoplastic drugs associated with CIPN is platinum-based drugs, such as oxaliplatin, which is frequently implicated in peripheral neurotoxicity and leads to the triggering of abnormal sensations like tingling, numbness, pressure, persistent pain, and heightened temperature sensitivity, causing significant suffering [13]. These symptoms usually manifest during the second or third treatment cycle and last for 2–4 days after the drug infusion. The exact mechanisms underlying CIPN remain incompletely understood, and practical strategies for its prevention and treatment remain unresolved challenges in medicine [14]. Nevertheless, numerous drugs have been employed over the past few decades to intervene in CIPN, and their effects have been evaluated. In addition, various compounds have been developed to prevent or treat CIPN by targeting ion channels, inflammatory cytokines, and oxidative stress [13].

The renin-angiotensin system (RAS) is a crucial player in regulating blood pressure and fluid homeostasis [15]. The octapeptide, Angiotensin II, is the first significant end product. Angiotensin II binds strongly to G-coupled protein receptors (GPCRs), Angiotensin receptor type 1 (AT1), to produce vasoconstriction, sodium-water retention, inflammation, and fibrosis; and it binds weakly to Angiotensin II receptor type II (AT2), which produces vasodilation and angiogenesis [16]. Angiotensin II is cleaved by Angiotensin-converting enzyme 2 (ACE2) to yield the heptapeptide, Ang-(1-7), which has a greater affinity for AT2 than AT1 [17]. Ang-(1-7), a biologically active heptapeptide, binds to the GPCR, Mas receptor (MasR1; Kd = 0.83 nM), with 60- to 100-fold greater selectivity over the AT1 and AT2 receptors, respectively [18,19].

Our team has taken a novel approach to treating CIPN by taking advantage of our extensive experience with the G-protein linked Mas receptor and its agonist Ang-(1-7) to develop a platform of glycosylated Ang-(1-7) Mas receptor agonists including PNA5 and PNA6 that have outstanding brain penetration, enhanced bioavailability, decreased brain and peripheral inflammation, improved cerebral blood flow, and restored cognitive function in our preclinical vascular dementia and TBI animal models. Activation of Mas decreases ROS and brain inflammation, increases cerebral circulation, increases induction of neuroprotective cytokines, and inhibits hypoxia-inducing factor-1alpha (HIF-1alpha) [20,21,22,23]. However, Ang-(1-7) is degraded rapidly by enzymes and has a short half-life in vivo. On the other hand, the glycopeptide PNA6, an O-linked lactoside analogue of Ang-(1-7), has been shown to have a half-life of over 1.5 h and good penetration into the CNS.

Permanent pain relief has not been achieved with the current therapies. Cancer-induced bone pain has become an increasing challenge, as new treatments are needed to maximize effectiveness while reducing adverse events. In the present study, we provide evidence that PNA6 may be especially effective in attenuating pain induced by inflammation, reversing CIPN, and relieving CIBP pain in mice.

## 2. Results

### 2.1. Solid Phase Peptide Synthesis (SPPS)

PNA6 was successfully synthesized and characterized; Figure 1 shows the chemical structure. Preparative HPLC was used to purify the crude, producing one gram of compound with 100% purity (at 280 m), confirmed by analytical HPLC as shown in Figure 2; the retention time of PNA6 is 7.59 min. The chemical composition and molecular mass of PNA6 were evaluated after purification using mass spectroscopy and showed the successful synthesis and purification of the targeted compound. The exact calculated mass for PNA6 (C_51_H_81_N_13_O_21_) was 1211.567 Dalton, and the found mass was 1211.62 Dalton, with [M + H] ^+^ 1212.35, and [M + 2H]^2+^ 606.81. Figure 3 shows the mass spectra for PNA6.

### 2.2. PK Analysis of PNA6 in Serum and Brain Parenchyma

To directly compare the in vivo lifetime and the ability of PNA6 to cross the blood–brain barrier (BBB), we devised a methodology that combines simultaneous blood sampling with microdialysis. PNA6 was administered via a single tail vein injection at a dosage of 10 mg/kg. Blood samples were collected at specific time intervals relative to the injection, including t = −10, 1, 5, 10, 20, 30, 40, 50, 60, 70, 80, and 90 min (with t = 0 corresponding to the moment of injection) as shown in Figure 4. The determination of serum and cerebrospinal fluid (CSF) concentrations of PNA6 involved the utilization of a calibration curve, which accounted for factors such as dilution and probe recovery specific to each experimental condition.

### 2.3. PNA6 Does Not Alter Cell Viability In Vitro

The concentration- and time-response curves for PNA6 on cell viability as compared to vehicle were assessed using an XTT assay. E0771 breast cancer cells were treated in vitro with different concentrations of PNA6 (0.01, 0.1, 1, and 10 µM) or serum-free media for 24 h to investigate whether the different concentrations of PNA6 alter tumor viability. In addition, an XTT assay was performed (Figure 5A). PNA6 did not significantly decrease the E0771 cell viability at any of the concentrations tested compared to the control cell viability (E0771+ DMEM). To investigate the effect of PNA6 on cancer cell viability at different time points, 96-well plates with E0771 cells were treated with 1µM of PNA6 and cell viability was measured at 1, 2, 3, and 4 h using the XTT assay. Figure 5B demonstrates that PNA6 did not significantly change the cell viability at multiple time points. This data indicates that PNA6 at the various concentrations tested and at acute time courses does not promote nor reduce cancer cell viability in vitro.

### 2.4. Effect of PNA6 on Carrageenan-Induced Mechanical Allodynia In Mice

The effect of PNA6 on acute inflammatory pain [24] was assessed after an i.paw. injection of 2% λ-carrageenan (Figure 6). An injection of 2% λ-carrageenan, but not saline, into the right hind paw caused increased sensitivity to tactile stimuli 3 h post-administration (Figure 6B,C). Hind-paw withdrawal thresholds in mice decreased from 2.097 +/− 0.146 SEM (baseline) to 0.462 +/− 0.073 SEM (n = 12) at 3 h post 2% λ-carrageenan administration, indicating that a significant mechanical hypersensitivity had developed (Figure 6C). Mice receiving the saline intra-hind paw as a control for the 2% λ-carrageenan and then given either saline (i.p.) or PNA6 (1 mg/kg, i.p.) had a decrease in thresholds from 1.922 +/− 0.215 SEM (baseline) to 1.456 +/− 0.262 SEM (n = 10) at 3 h post-saline, most likely due to the needle injection. Still, they were not significant between the saline and PNA6 treatment groups. PNA6 in control animals did not significantly increase or decrease mechanical thresholds as compared to the saline-treated mice (Figure 6B). A single systemic intraperitoneal administration of PNA6 (1 mg/kg) or vehicle (saline) was administered in 2% λ-carrageenan animals, and pain behavior using von Frey was assessed on the ipsilateral hind paw. A time-response curve of PNA6 in an acute inflammatory model demonstrated a significant anti-mechanical hypersensitivity (* *p* < 0.05, ** *p* < 0.01, n = 12) at the 60, 90, and 120 min time points but not at the 30 nor the 180 min time points as compared to the saline control group (Figure 6). These data suggest that PNA6 effectively mitigates acute inflammation.

### 2.5. Chronic Administration of PNA6 Significantly Attenuates Pain in a Murine Chemotherapy-Induced Peripheral Neuropathy Model (CIPN)

To exclude the effect of PNA6 on motor coordination, we injected mice with either vehicle or PNA6 (1 mg/kg i.p.) systemically, and motor impairment was assessed using the rotarod test before drug administration and re-evaluated at 30, 60, 90, and 120 min after treatment (Figure 7A). The data showed that PNA6 neither increased nor decreased the time that naïve mice spent on the rotating rod compared to vehicle-treated mice (Figure 7B).

To study the effect on chemotherapy-induced peripheral neuropathic pain, PNA6 was administered 14 days after inducing peripheral neuropathy pain in mice (Figure 7A). Baseline mechanical thresholds using von Frey filaments were recorded on the right hind paw of the mice (4.000 +/− 000 SEM). Next, after 30 min of oxaliplatin injection, PNA6 (1 mg/kg, i.p.) or vehicle (saline, i.p.) was administered, and mechanical hypersensitivity was reassessed utilizing von Frey filaments on the right hind paw (1 hr post admin), depending on the maximum effect of PNA6 shown in the time-response curve on day 4 and day 11 (Appendix A). The oxaliplatin-PNA6 treatment group showed a significant reversal in the paw withdrawal thresholds (3.280 +/− 0.389 SEM) as compared to vehicle-treated animals (0.635 +/− 0.180 SEM) (Figure 7C). The area under the curve was calculated to assess the time response (Figure 7D). Together, these data suggest that administration of PNA6 significantly reduces mechanically evoked pain responses in an established mouse CIPN model.

### 2.6. Acute Administration of PNA6 Attenuates Cancer-Induced Bone Pain

We evaluated the antinociceptive efficacy of PNA6 in an established murine model of cancer-induced bone pain derived from female murine metastatic breast cancer [11]. In this syngeneic model, we implanted E0771 breast adenocarcinoma cells into the medullary canal of the right femurs of healthy C57 black/6J WT female mice. Before cancer inoculation of the femur, the mice did not display behavioral signs of pain including mechanical allodynia, flinching, or guarding. Seven days after intramedullary implantation of E0771 cancer cells or media only (no cancer cells as a sham control), flinching, guarding, and limb use were re-observed to record the cancer-induced bone pain (CIBP) baseline. Mice implanted with E0771 adenocarcinoma cells showed a significant induction of pain behaviors as compared to the sham mice (mice injected with media only). The cancer-treated mice that received PNA6 (1 mg/kg, i.p.) showed a significantly reduced number of flinches compared to saline-treated mice at 60 min post administration (Figure 8A); although flinching decreased compared to saline at the additional time points (30, 90, and 120 min), statistical significance was not reached. The sham-treated mice that received PNA6 (1 mg/kg, i.p.) were not statistically distinguishable from saline-treated animals. Likewise, in the behavior of cancer-induced guarding, the cancer-treated mice that received PNA6 (1 mg/kg, i.p.) significantly reduced the guarding at the 60 min time point compared to saline-treated mice (Figure 8B). The sham-treated mice that received PNA6 (1 mg/kg, i.p.) were no different from saline-treated animals. Limb-use behavior was reduced in the cancer-treated mice and this was normalized in mice that received PNA6 (1 mg/kg, i.p.) at the 60 and 90 min time points compared to saline-treated mice (Figure 8C). The sham-treated mice that received PNA6 (1 mg/kg, i.p.) were no different from saline-treated animals with respect to limb use. 

### 2.7. Chronic Administration of PNA6 Significantly Attenuates Cancer-Induced Bone Pain (CIBP) in a Murine Model of Metastatic Breast Cancer

To assess the effect of chronic exposure to PNA6 against cancer pain, we separated cancer and sham mice into two groups. We injected PNA6 1 mg/kg, i.p., once daily for seven consecutive days from day 7 to day 13 after surgery. Behavior tests of flinching, guarding, and limb use were observed on days 7, 10, and 13 post-surgeries at 60 min based on the acute study. Mice inoculated with cancer cells significantly increased flinching and guarding, and decreasing limb use on day 7 (Figure 9). Sham animals (surgery but no cancer inoculation) also showed pain behaviors, but significantly lower than the cancer-inoculated group. PNA6 significantly lowered the flinching and guarding behaviors in the cancer-saline group on days 7, 10, and 13 (Figure 9A,B) compared to the cancer-inoculated animals that received sustained saline from days 7 to 13. The limb-use test showed that PNA6 injection significantly increased the limb-use score in mice with cancer on days 7 and 10, but was insignificant on day 13 (Figure 9C) compared to the saline-treated animals. Neither PNA6 nor saline had any significant effect in sham-inoculated animals. 

To explore whether PNA6 elicits its antinociceptive effects on CIBP in vivo through a MasR1 dependent mechanism, A779 a MasR1antagonist was used. Cancer-bearing mice were administered either the MasR1 antagonist A-779 (1 mg/kg) or vehicle 30 min before PNA6 (1 mg/kg) for seven consecutive days. The MasR1 antagonist, A-779, alone did not change spontaneous pain behavior in mice with CIBP. Pretreatment with A779 significantly inhibited PNA6 attenuation of flinching (Figure 9D), guarding (Figure 9E), and limb use (Figure 9F). Together with the acute dosing paradigm on day 7, these data indicate that the chronic administration of PNA6 alleviates behaviors associated with spontaneous pain in metastatic bone disease with retained efficacy by targeting the MasR1.

### 2.8. Sustained Administration of PNA6 Did Not Alter Bone Integrity of Mice with Established Cancer-Induced Bone Pain (CIBP)

Radiographic images of mice’s right femurs with established (CIBP) or sham were captured using a Faxitron machine. Bone loss was rated and evaluated using a bone score scale by a blinded observer where 0 = normal healthy bone; 1 = 1–3 lesions; 2 = 4 to 6 lesions; 3 = uni-cortical bone fractures; and 4 = bicortical bone fractures. Images of the ipsilateral femur on day 14, comparable to day 0, showed that all mice injected with media showed no or mild bone loss. Sustained saline or PNA6 had no change in bone remodeling in sham animals. While almost all cancer-bearing mice treated with sustained saline or PNA6 experienced severe bone loss with a score of 2 or more (Figure 10), chronic treatment with PNA6 did not exacerbate cancer-induced bone loss degradation compared to the vehicle group.

## 3. Discussion 

Despite the improvement of therapeutic techniques to reduce cancer mortality, including chemotherapy, radiotherapy, surgery, and bio-chemotherapy, no new developments have targeted the number one complaint, chronic pain and chemotherapy-induced neuropathic pain. Recent studies demonstrate RAS’s role in the development and invasiveness of several types of cancers. Ang-(1-7), an essential component of the RAS system, demonstrated an implication in metastatic disease and pain [23,25,26], yet no studies have investigated whether PNA6/MasR1 was active in murine models of inflammation, CIBP or CIPN. Evidence shows that mRNA and protein expression of the Mas receptor (MasR1) is upregulated in DRG after chronic constriction injury to the sciatic nerve, and Ang-(1-7) significantly inhibits the neuropathic pain in a chronic constriction injury [27]. Since cancer-induced bone pain (CIBP) and chemotherapy-induced neuropathy display characteristics of on-going neuropathic and inflammatory pain [10], we chose to investigate the utility of PNA6, a Lactosyl Ang-(1-7) analog with a longer half-life and good CNS penetration, in murine models of inflammatory and neuropathic pain. We tested both acute and chronic systemic administration of PNA6 to attenuate the spontaneous pain behaviors induced by carrageenan-induced inflammation, oxaliplatin-induced peripheral neuropathy in mice, and breast cancer confined to the long bones in murine models of pain.

Native peptides are known for their inherent instability in vivo, with short half-lives of the biologically active peptide and low permeability across biological membranes. Therefore, several strategies have been applied to improve the native peptides’ stability, safety, circulation time, efficacy, biodistribution, etc. These strategies are PEGylation, lipidation, conjugation to antibodies, and stapled peptides [28]. One promising strategy to synthesize our compound of interest here (PNA6) is glycosylation. Glycosylation is a synthetic method for introducing carbohydrate moieties to peptides [29]. Some advantages of peptide glycosylation include protecting the amino acids’ side chain from oxidation, increasing metabolic stability, enhancing the penetration through biological membranes, targeting specific organs, improving the biodistribution in tissues, lowering the clearance rate, enhancing receptor binding, and stabilizing the physical properties of peptides [30,31,32,33]. As Ang-(1-7) is a native bioactive peptide with all the above-mentioned therapeutic potential, our group developed derivatives of it via the glycosylation strategy. One of these derivatives is PNA6, successfully synthesized using the SPPS strategy [34] with a high purity from the glycosylation of Ang-(1-7) (DRVYIHP) replacing the seventh residue (proline) with serine. A lactoside was attached to the hydroxyl group of the serine, and the C-terminus was acetamidate, resulting in Ang-1-6-Ser-(β-O-Lact)-CONH2 (PNA6). This synthesis approach is like the one performed to synthesize the ser-glucoside derivative of Ang-(1-7) (PNA5) [35]. PNA6 is an amphipathic glycopeptide that contains a carbohydrate moiety. As mentioned above, the sugar moiety affects its interaction with the biological membrane and increases its aqueous solubility. In general, glycopeptides have relatively lipophilic backbones and hydrophilic sugar moieties; this allows the O-glycopeptide to interact with both the membrane surface and the aqueous compartment in a “hopping” motion, which promotes the transport of the drug throughout the body in vivo, improves bioavailability, shows more stability, less degradation by enzymes, and improves the half-life [29,36]. The modification of peptides demonstrates potential in the design of novel therapeutics that tend to have high efficacy while lacking unwanted side effects [37].

Our findings demonstrating that PNA6 had long-acting efficacy when given acutely as well as chronically suggest that a modified natural peptide may be helpful in reducing CIBP. Since some drugs can cause paralysis or decrease limb use resulting in the misinterpretation of antinociceptive effects, we demonstrated that PNA6 had no significant effect on the motor movement of the limbs using a rotarod. Our data indicate that the mechanism of action through which PNA6 inhibits pain involves its interaction with the MasR1, a G-protein coupled receptor, within either the tumor-peripheral nociceptor and/or the CNS. Our data indicate that PNA6 directly targets nociceptive activation rather than exerting its effects through the tumor–bone environment. This is supported by the finding that PNA6 did not significantly alter tumor-induced bone degradation, as demonstrated by many compounds that alter such an environment and have an influence on osteoclast activity [38]. Unlike compounds that modulate bone wasting and may influence pain through osteoclast activity, our chronic studies with PNA6 did not result in a significant bone loss. An additional explanation of PNA6′s ability to inhibit CIBP would be the potential slowing of tumor proliferation, yet the chronic administration of PNA6 did not have a substantial impact on the tumor burden, further suggesting that its analgesic effect is primarily mediated by inhibiting nociceptive activation. These findings shed light on the specific mechanism by which PNA6 attenuates pain and highlight its potential as a targeted therapeutic approach for pain management in the context of the tumor–nociceptor microenvironment.

One may anticipate alterations in MasR1 expression within the nociceptive circuit and/or the bone–tumor microenvironment if MasR1 is a viable therapeutic target for CIBP and the site of action for PNA6. Our previous data confirm that MasR1 is expressed in the DRG neuron of naive mice, and cancer cell inoculation significantly increased the expression of MasR1 in the ipsilateral femur extrudate, in accordance with a previous report of neuropathic pain. The study utilized a rat model of chronic nerve injury to demonstrate that the MasR1 expression in DRG neurons is induced over time [39]. The increased MasR1 expression was observed at both the mRNA and protein levels, leading to enhanced MasR1–ligand binding on DRG neuron membranes. Our behavioral tests confirmed that MASR1 activation improved pain symptoms, while MASR1 inhibition attenuated the effects of PNA6.

Metastatic cancer is most often treated with chemotherapy [40]. Such types of treatments are becoming more successful with patients living longer but with more unwanted side effects induced by the treatment. One such side effect that can often result in the diminution of the treatment is chemotherapy-induced peripheral neuropathy (CIPN) [41]. CIPN is a debilitating and persistent side effect of several antineoplastic agents that often results in sensory abnormalities such as tingling, numbness, pressure, persistent pain, and thermal pain hyperalgesia [42]. The pathogenesis of CIPN remains largely unknown, and its prevention and treatment are ongoing challenges in medicine. CIPN is commonly associated with platinum-based drugs such as oxaliplatin [43]. Numerous drugs have been developed and evaluated over the years to try and prevent or treat CIPN, including targeting ion channels, inflammatory cytokines, and oxidative stress, yet no new treatment is available for CIPN.

The idea of significantly attenuating CIPN by a pretreatment with PNA6 to reduce the incidence and/or severity of CIPN would be optimal and allow for proper chemotherapy dosing to overcome metastatic cancer.

Research on human neuropathic pain and animal models has consistently shown that chemotherapeutic agents lead to the activation of glial cells, triggering the release of pro-inflammatory cytokines (e.g., IL-1b, IL-6, and TNF-a) [44] and chemokines (e.g., CCLs, CX3CL1, and CXCL12) [45]. This cytokine release is linked to Toll-like receptor activation, particularly TLR4, as is evident from the reduced pain behavior in TLR4 knockout mice treated with cisplatin [46]. Interestingly, a recent study identified this same target, TLR4, as a mechanism of chronic morphine-induced pain in a murine model of bone cancer pain [38], potentially limiting opioids as a treatment for CIBP. Pro-inflammatory cytokines have been shown to sensitize nociceptors by modulating ion channel properties, as evidenced by the research conducted by Jin et al. [47]. Oxaliplatin can increase the level of the CCL-2 chemokine, primarily released from astrocytes, and the level of CCL-2 is correlated with the degree of hyperalgesia observed in rats [48]. Other studies have confirmed the role of chemokine CXCL12 [49] and chemokine CX3CL1 signaling in oxaliplatin-induced neuropathy [50].

While previous research has highlighted the anti-inflammatory effects of Ang-(1-7) mediated through MasR1 activation, its potential role in CIPN treatment remains unexplored. Ang-(1-7) has been shown to significantly reduce proinflammatory cytokines (TNF-α, IFN-γ, IL-1β, and IL-6) and promote the production of the anti-inflammatory cytokine IL-10, contributing to overall inflammation suppression [51,52].

In conclusion, our findings suggest that PNA6 may be a promising therapeutic agent for the treatment of CIBP and CIPN. Notably, PNA6 was effective in mitigating pain without altering tumor growth or bone loss, unlike what has been reported with the use of opioids, which is the current clinical practice [38]. This indicates that patients with bone cancer pain may still receive chemotherapeutic intervention, which is often halted or reduced due to CIPN, while being treated with PNA6, further eradicating the metastasis proliferation while having pain relief. Further studies are needed to evaluate the efficacy of PNA6 in human subjects and to further evaluate its mechanism of action in treating CIPN. Overall, our results suggest that PNA6 may have therapeutic potential for managing CIPN, a significant challenge in oncology care. Moving forward, further investigations are warranted to unravel the precise molecular pathways through which PNA6 exerts its antinociceptive effects. Understanding the specific inflammatory and pain pathways modulated by PNA6 will provide valuable insights for the development of novel therapeutic strategies for pain management in cancer patients. Additionally, the potential of PNA6 or similar glycosylated derivatives as a viable treatment option for neuronal damage and PNS to CNS inflammation warrants comprehensive exploration in preclinical and clinical studies. These advancements hold promise for addressing the unmet needs of cancer patients suffering from chronic and neuropathic pain, ultimately improving their quality of life during their treatment.

## 4. Materials and Methods 

### 4.1. Solid Phase Peptide Synthesis (SPPS) 

The reagents used for SPPS were Rink amide MBHA resin (100−200 mesh, 0.83 mmol/g), 6-chloro-1-hydroxybenzotriazole (Cl-HOBt), Fmoc-L-His(Trt)-OH, Fmoc-L-Ile-OH, Fmoc-L-Tyr(tBu)-OH, Fmoc-L-Val-OH, Fmoc-L-Arg(Pbf)-OH, *N*, *N*-Methylmorpholine (NMM), and 1,8-Diazabicyclo [5.4.0]undec-7-ene (DBU), and they were obtained from Chem-Impex INT’L INC (Wood Dale, IL, USA). Fmoc-Pro-OH, Fmoc-Asp (OtBu)-OH, and 2-(1H-benzotriazol-1-yl)-1,1,3,3-tetramethyluronium hexafluorophosphate (HBTU), and *N*, *N*-diisopropylcarbodiimide (DIC), as well as Fmoc-L-Ser(β-O-LactAc7)-OH, were obtained from AAPPTec (Louisville, KY, USA). Anisole was obtained from Aldrich Chemical CO., INC (Milwaukee, WI, USA). N, N-Diisopropylethylamine (DIPEA) was obtained from Sigma-Aldrich, Co (St. Louis, MO, USA). *N*-methyl pyrrolidone (NMP) was obtained from VWR Chemicals BDH^®®^ (Radnor, PA, USA). Dichloromethane (DCM) was obtained from Fisher Chemical (Geel, Belgium). N, N-Dimethylformamide (DMF). Piperidine and Hydrazine monohydrates were obtained from Alfa Aesar (Ward Hill, MA). Trifluoroacetic acid (TFA) and Triethylsilane (TES) were obtained from Oakwood Chemical (Columbia, SC, USA). Ether anhydrous was obtained from Avantor–J.T.Baker^®®^ (Radnor, PA, USA).

The PNA6 drug candidate (Asp-Arg-Val-Tyr-Ile-His-Ser(β-O-Lact)) was assembled using standard Fmoc-based solid-phase peptide synthesis on Rink amide-MBHA resin (loading: 0.83 mol/g). First, the coupling of Fmoc-Ser(β-O-LactAc_7_)-OH (1.3 eq) was completed manually using Cl-HOBt (1.3 eq) and DIC (1.3 eq) in NMP, capping of unused sites on the resin was accomplished with 10% acetic anhydride and 10% DIPEA in DCM. Then, the coupling of the Fmoc-protected amino acids was performed on a Prelude^TM^ automated synthesizer. Coupling was accomplished with 2.5 eq Fmoc-AA and HBTU (3.0 eq) and 12 eq of NMM in DMF.

The Fmoc groups were removed using a mixture of 2% piperidine and 2% DBU in DMF for 12 min with argon bubbling as agitation. Next, the acetyl-protecting groups of the sugar moiety were removed by a 1:1 mixture of hydrazine monohydrate in NMP (H_2_NNH_2_•H_2_O: NMP) with argon agitation for 4 h. Finally, the PNA6 was cleaved from the resin with a peptide cleavage cocktail consisting of 89% TFA, 10% DCM, 0.25% TES, 0.25% water, and 0.5% Anisole for 1 hr at room temperature, which simultaneously removed the side chain protecting groups.

For the HPLC purification and characterization of PNA6, the crude was precipitated in cold ether, dissolved in a minimal amount of distilled water, and then lyophilized. First, the crude material was purified by RP-HPLC Gilson system on a semi-preparative C_18_ Phenomenex column (5 µm, 100 Å, 250 mm × 50 mm) using gradient program mobile phase of (A: 5–80% ACN) vs. (B: 0.1% TFA in H_2_O) over 60 min to give the compound in a pure form. Then, the purity was assessed using analytical HPLC (Inspire C18 5 μm 250 mm × 4.6 mm column) on a Varian LC with a diode array detector system (at 280 nm), employing the same mobile gradient phase program for 15 min. The pure fractions obtained from preparative HPLC purification were frozen at −80 °C and then lyophilized to afford pure PNA6 as a white and fluffy solid. The same analytical RP-HPLC was used to evaluate the purity of the compound after lyophilization. Finally, the mass measurements of pure synthetic PNA6 were performed and characterized using mass spectrometry (Bruker AmaZon ion trap mass spectrometer via infusion, Analytical and Biological Mass Spectrometry, Core Facility in BIO5 Keating Bioresearch Building, University of Arizona, Tucson, AZ, USA).

### 4.2. In Vitro 

#### 4.2.1. Cell Culture 

E0771 breast adenocarcinoma cells were gifted by Kathryn Visser at the Olivia Newton-John Cancer Research Institute Metastasis Research Institute (Victoria, Australia). We injected 8 × 10^4^ E0771 breast adenocarcinoma cells (P10–20) suspended in 5 uL volume of OPTI-MEM into the medullary cavity into the right femur of female C57BL/6J wild-type mice to induce cancer-induced bone pain (CIBP). Cancer cells were cultured in DMEM—Dulbecco’s Modified Eagle Medium—with 10% fetal bovine serum (FBS), 1% penicillin /streptomycin (P/S), and 25 mM HEPES buffer. The E0771 cells were plated in T-75 tissue culture flasks and allowed to grow exponentially in an incubator at 37 °C and 5% CO_2_. Cells were passaged every four days or as needed, and cells used for all experiments were in the 10–20 passage range.

#### 4.2.2. Cell Viability Assay

We assessed the effect of PNA6 on the viability and performance of the E0771 adenocarcinoma cells. First, E0771 adenocarcinoma cancer cells were thawed and resuspended in growth media, then plated in T-75 tissue culture flasks, allowing them to grow in the incubator at 37 °C with 5% CO_2_. Next, the cancer cells were inoculated in a flat-bottom 96-well microtiter plate at a concertation of 1 × 10^6^/ well in serum media (determined from the pre-optimized procedure) and placed in the incubator for 24 h. After 24 h, the media was removed using a suction pipette; the cells were treated with respective pharmacological treatments in serum-free media. The treatments utilized were serum-free media alone, cells with media, different doses of PNA6 (0.01, 0.1, 1, and 10 µM) for another 24 h. Cell viability to the different drug doses were assessed after 24 h with the tetrazolium salt XTT cell proliferation assay according to the manufacturer’s protocol (catalog number 30–1011K; ATCC, Manassas, VA, USA).

### 4.3. In Vivo

#### 4.3.1. Animals

All procedures were approved by the University of Arizona Animal Care and Use Committee and conformed to the National Institutes of Health Guidelines and the International Association for the Study of Pain (Protocol 19–600, Approved 19 July 2021). Male CD1 mice (n = 44) 18–20 g (7–8 weeks) were used for the inflammatory model study. Adult female C57BL/6J wild-type mice between 15 and 20 g (7–8 weeks) were used in the cancer study (n = 10–12 mice per treatment group) for the syngeneic implant of E0771 cells. In addition, 24 adult female C57BL/6J wild-type mice were used in the CIPN study. Mice were housed in a climate-controlled room on a 12 h light–dark cycle and allowed food and water ad libitum. Animals were monitored on days 0, 7, 10, and 13 of the study for clinical signs of rapid weight loss (less than 20% in 1 week) and anxiety symptoms. All mice were euthanized utilizing an IACUC-approved CO_2_ procedure.

Animals for Dialysis: Male Sprague Dawley rats (275–325 g) were used for dialysis experiments in vivo. Animals were purchased from Envigo Laboratories (Indianapolis, IN, USA) and housed in a temperature and humidity-controlled room with 12 h reversed light/dark cycles with food and water available ad libitum. All animals were treated as approved by the Institutional Animal Care and Use Committee, University of Arizona, and in accordance with the NIH Guidelines for the Care and Use of Laboratory Animals. Both the number of animals used and their suffering were minimized. The number of animals needed for each experiment was determined using a G*Power analysis (Faul and Coronado).

#### 4.3.2. Drug Treatment

Animals received PNA6, a lactosyl Ang-(1-7) analog, or vehicle (saline) in the absence or presence of the MasR1 antagonist A-779 (Abcam, Cambridge, MA, USA). All injections were made by the intraperitoneal (i.p.) route at a dose of 1 mg/kg (10 mL/kg). In the antagonist studies, A-779 (1 mg/kg, i.p.) was administered 30 min before PNA6 administration. Therefore, only one PNA6 injection (1 mg/kg i.p.) or vehicle (saline) was given on day seven of post-cancer cells femur inoculation for the acute inflammatory study. For the chronic study, administration of PNA6 or vehicle (saline) was started on day seven post-cancer cell femur inoculation and continued with once-daily injections of PNA6 (1 mg/kg, i.p) or vehicle for seven consecutive days. Dosing was performed regularly at 9:00 am. Lastly, for the CIPN study, oxaliplatin (4 mg/kg i.p.) (Tocris Bioscience a Bio-Techne Brand, 61825-94-4, catalog No.: 2623) was prepared daily in a vehicle of sterile 5% dextrose solution distilled water. Systemic administration of PNA6 (1 mg/kg, i.p) or vehicle was started on day one and continued with once-daily injection for 14 consecutive days. 

#### 4.3.3. Carotid Artery Catheterization

Catheterized animals were purchased directly from Envigo Laboratories (Indianapolis, IN). At their facility, animals were anesthetized with 1.5–2% isoflurane mixed in medical grade oxygen (1.5 L/minute), and two incisions were made: (1) a 0.5 cm midline incision between the scapulae; and (2) a 2.0 cm incision to the right of the midline of the ventral neck. The right carotid artery was isolated, and two 4-0 silk sutures were placed at the caudal and cranial ends of the vessel. The cranial suture was immediately tied off and a bulldog clamp was placed above the caudal tie to temporarily occlude blood flow. A carotid venotomy was performed and a piece of PE-10 catheter tubing, pre-flushed with a heparin:glycerol (500 IU Heparin/1 mL; 50:50 ratio) solution, was passed through the vessel towards the heart. The bulldog clamp was then released, and the caudal suture was tied off to secure the catheter in place. The catheter was then threaded through a subcutaneous pocket and connected to a vascular access port (VAP; Instech Laboratories, Plymouth Meeting, PA, USA) at the scapular incision. The ventral neck incision was closed with stainless steel wound clips, while the VAP was secured, and the scapular incision was closed with 4-0 silk sutures. Animals were monitored during recovery for 3–4 days before being shipped to University of Arizona facilities. Catheter patency was evaluated within the first 24 h of arrival and subsequently, every 2–3 days to ensure patency prior to microdialysis. 

#### 4.3.4. Microdialysis Surgery

Upon arrival at our facility, animals were acclimated for 5–7 days. Procedures for microdialysis have been described previously (Heien; Mabrouk et al.). Animals were anesthetized with 1.5–2% isoflurane mixed in medical grade O_2_ (1.5 L/minute) and positioned inside a stereotaxic frame (David Kopf Instruments, Tujunga, CA, USA). A midsagital incision was made along the skull and a CMA 11 guide cannula (CMA Microdialysis, Kista, Sweden) was implanted in the striatum at the following coordinates: AP, +1.0 mm, and ML, +3.2 mm, relative to bregma and DV −3.4 mm from the surface of the brain. The guide was then replaced with a CMA 11 probe containing a 4.5 mm cuprophane (Cupr) membrane with a molecular weight cut-off of 6 kDa. After the probe was positioned, the animals remained under anesthesia for the remainder of the experiment. 

#### 4.3.5. Microdialysis and Blood Draws

The compounds were injected intravenously at 10 mg/kg via a single tail vein injection. Blood draws were taken at t = −10, 1, 5, 10, 20, 30, 40, 50, 60, 70, 80, and 90 min (where t = 0 corresponds to injection). Microdialysis fractions were time-locked with blood draws at 10 min increments such that blood draws correlated with the median time of the microdialysis fraction. For microdialysis, a dual syringe pump (PHD 2000, Harvard Apparatus, Holliston, MA, USA) was set to a flow rate of 0.5 mL/minute. Samples were collected and preserved in a solution containing 10% *v*/*v* acetic acid, 2% *v*/*v* acetonitrile, and 140 pM D-alanine D-leucine enkephalin (DADLE). DADLE was used as an internal standard (IS). Baseline blood (~100 mL) and dialysate samples were collected. Subsequent time-locked samples (with blood being drawn at the median of the dialysate time range) were then collected every 10 min for 90 min, following a tail vein injection at t = 0 min. In addition, blood samples were also collected at both 1 and 5 min post-injection. CSF samples were immediately frozen on dry ice. Blood samples were centrifuged to separate serum from the red blood cells. Serum was drawn off, and 10 mL of serum was spiked with glacial acetic acid (HOAc) to quench peptidase activity, and DADLE was added to a final concentration of 5% HOAc and 140 pM IS before being frozen on dry ice. Post-experiment, samples were stored in a –80 °C freezer until analysis.

All samples were desalted using µ-C_18_ Zip Tips^®®^ (EMD Millipore) according to the protocol above, yielding a final solution comprising 10 µL 60:40 ACN:H_2_O:0.1% TFA (*v*/*v*). For reverse-phase liquid chromatography, samples should be dissolved in aqueous solution, thus samples were vacuum-centrifuged until 1 µL of solution remained and reconstituted in 5 µL H_2_O with 0.1% TFA (*v*/*v*). Samples (5 µL) were injected into the column and separated using a gradient elution. Target compounds were quantified by summing the height of previously identified fragment peaks in the LC-MS^3^ spectra. The serum and CSF concentrations were estimated using a calibration curve, accounting for the dilution factors and the probe recoveries for each experiment. Standards were matrix-matched to samples and underwent the same sample preparation steps. Data Analysis: all data are presented as mean ± standard error (SEM). 

#### 4.3.6. Induction of Inflammatory Model of Pain

Behavioral assays (mechanical allodynia) using von Frey testing were performed as a baseline measurement of mechanical thresholds of the right hind-paw (details described below). Male CD1 mice (n = 40) were anesthetized using nose-cone isoflurane/O_2,_ and acute inflammatory pain was induced by injecting 50 μL of 2% λ-carrageenan or saline subcutaneously (s.c.) into the plantar surface of the right paw. After 3 h, redness and swelling signs and symptoms of inflammation appeared in the right paw ipsilateral to the site of 2% λ-carrageenan injection. Mechanical allodynia was reassessed via application of von Frey filaments, which was performed 3 h after a 2% λ-carrageenan injection. Mice were then injected with PNA6 (1 mg/kg, i.p.) or vehicle (i.p.). Mechanical allodynia testing was carried out in a blinded-to-treatment manner, at intervals of every 30 min (30, 60, 90, 120, and 180 min) post-drug or vehicle administration over a 3 h time course or until their pain behaviors returned to baseline. 

#### 4.3.7. Intramedullary Implantation of E0771 Cells into Mouse Femur

Intramedullary implantation of E0771(murine breast cancer cells) into the right femur of C57BL/6J wild-type female mice was performed as previously described [53]. On day 0, mice were anesthetized with 80 mg/kg ketamine to 12 mg/kg xylazine (in a 10 mL/kg volume) i.p.. The area of the surgery (right hind leg) was shaved and then cleaned with antiseptic chlorhexidine three times, then the thigh muscle was exposed by lateral incision over the femur. An arthrotomy was performed, and the condyles of the right distal femur were exposed. A hole was drilled at the intercondylar space to access the medullary cavity of the right distal femur. A total of 80,000 E0771 cells were suspended in 5µL volume of OPTI-MEM, then breast adenocarcinoma cells were injected into the medullary cavity using an injection cannula affixed via plastic tubing to a 10 µL Hamilton syringe; sham animals received only OPTI-MEM. The femur was then sealed using bone cement, and the muscular compartment was closed using a 5-0 vinyl suture. The skin was closed using wound clips. Gentamicin (8 mg/kg, 10 mL/kg volume) and normal saline (3 mL, s.c.) were given to all mice after surgery to prevent infection and dehydration. The mice were monitored on D1, D2, and D3 to check the presence of clips and monitored clinical signs of paralysis. Animals were allowed seven days to recover, and clips were removed. Any animal showing signs of illness (loss of significant weight, lack of regular motor movement, etc.) or animals with radiographic signs of fracture before day 14 post-surgery were removed from the study, and their data were not included in the analysis.

#### 4.3.8. Induction of Chemotherapy-Induced Peripheral Neuropathy In Mice

Oxaliplatin (4 mg/ kg) was prepared daily in a vehicle of sterile 5% dextrose solution. Animals received systemic intraperitoneal (i.p.) injections of vehicle or dose (4 mg/kg-dose) oxaliplatin for two consecutive days, followed by five days of rest, followed by a second cycle of two daily i.p. injections, followed by five days of rest. The total cumulative doses of oxaliplatin over the course of four injections was 16 mg/kg. The dose of the drug was selected based on the literature evidence, which indicates its effectiveness in inducing neuropathic pain without unspecific systemic toxicity [54]. Moreover, the dose used in this study is relevant to clinical dosing. The highest recommended dose of oxaliplatin patients is 110 mg/m^2^ (2.97 mg/kg) [55]. The total human equivalent dose (THED) of oxaliplatin daily dosing ranges in mice is 0.04–10.0 mg/kg/day, and the cumulative is 3.0–30.0 mg/kg (0.24–2.4 mg/kg THED) [56]. Mechanical allodynia of the right hind paw was determined using manual von Frey filaments. After habituation to the test environment and baseline measurements of pain sensitivity, mice were randomized into two treatment groups of either PNA6 (1 mg/kg) or vehicle. Mice were treated with daily intraperitoneal (i.p.) injections for 14 consecutive days. Mechanical hypersensitivity thresholds were assessed every day at the same time for 14 days after 1 h of PNA6 or vehicle injection. Mechanical allodynia testing was carried out in a blinded-to-treatment manner.

#### 4.3.9. Measurement of Mechanical Allodynia

Mechanical allodynia was measured according to the method described previously [57]. Paw withdrawal threshold of the ipsilateral paw (to the injury site) to assess mechanical hypersensitivity utilized a series of calibrated von Frey filaments (0.04–4.0 g). Mice were tested for preinjury baseline mechanical hypersensitivity and after 3 h (baseline after injury), as well as after 30, 60,90, 120, and 180 min post PNA6 i.p. injection. The mice, after placement in a plexiglass cage with mesh metal flooring, were acclimated to the environment for 30 ± 5 min. After test cage acclimation, von Frey filaments of 0.16, 0.4, 0.6, 1, 1.4, and 2 g were applied perpendicularly at the plantar surface in the middle of the large right hind paw of the mice using the up and down method. Each filament was used with a force that caused slight bending and was held at the same force for 2–3 s. The filament of 0.6 g was applied as a starting filament to the paw of the mice kept in mesh metal flooring. In the absence of a hind paw withdrawal response to the selected filament, a higher force filament matching a stronger stimulus was applied; a filament of lighter force was chosen in the event of paw withdrawal response. The same procedure was repeated for five consecutive readings, and the average value of five scores was calculated using a Dixon nonparametric test and expressed as the mean withdrawal threshold. A 2 g force was selected as the cut-off force.

#### 4.3.10. Acute and Chronic Behavioral Testing of Spontaneous Pain

Flinching, guarding, and limb use are spontaneous pain-related behaviors that accurately measure ongoing pain in the cancer-bearing limb. Flinching is lifting and rapid flicking of the hind paw ipsilateral to the cancer inoculation limb not associated with walking or other movements. However, if the mouse shook its foot while walking, this was also counted as a flinch; this behavior test counted how many times the mice flinched over 2 min. Guarding is characterized by fully retracting the ipsilateral hind paw under the torso. Time spent guarding the tumor-bearing limb over a 2 min period was quantified using a stopwatch. Limb-use was tested by allowing mice to move freely in a plexiglass cage individually, while the gait of each mouse was observed and scored as follows: 4, normal use of the affected limb; 3, limping; 2, limp and guard; 1, partial non-use; and 0, complete lack of use of the affected limb. Mice were allowed to acclimate for 30 min before testing. All mice were baselined for these behaviors at day 0 before surgery. Animals were then re-examined for pain behaviors on days 7, 10, and 13 post-surgery. Behaviors were analyzed by treatment-blind examiners.

##### Acute Behavioral Testing

Each animal was observed for 2 min. Flinching, guarding, and limb use were evaluated at baseline day 0 (before surgery). Behavioral testing was repeated on day seven post-femur-cancer inoculation. Mice were separated into groups; each group had 10–12 mice and consisted of Sham–PNA6 (1 mg/kg), Sham–Saline, Cancer–PNA6 (1 mg/kg), and Cancer–Saline mice, receiving a single dose of PNA6 or saline. Spontaneous pain behavior was assessed at 30, 60, 90, and 120 min after administration of PNA6 (i.p.) or saline (i.p.). An acute time-response curve was generated on day 7. Behaviors were analyzed by treatment-blind examiners.

##### Chronic Behavioral Testing

Spontaneous pain behaviors were assessed before surgery (baseline) and on day seven post-surgery as a new injured CIBP baseline. Mice then received treatment at the same time each day (9:00 am ± 30 min) from days 7 to 13. Spontaneous pain behaviors were measured 1 h after treatment on days 7, 10, and 13 based on the time of peak effect determined by the acute studies. Mice were grouped as follows, Sham-PNA6 (1 mg/kg, i.p.), Sham-Saline, Cancer-PNA6 (1 mg/kg, i.p.), and Cancer-Saline, with 10–12 mice in each group. Behaviors were analyzed by treatment-blind examiners.

#### 4.3.11. The Rotarod Test

The rotarod test is widely used to evaluate the motor coordination of rodents, and we used it to determine the motor effect of PNA6 to prevent interpretation of behavior as a false positive. Four days before testing, naïve mice were trained to acclimate to the rotating rod (LE8505 Rota-Rod, Panlab Harvard Apparatus, Spain) at a constant speed of 10 rpm. A maximal cut-off time of 180 s was used to prevent exhaustion. On the day of testing, mice were baselined and re-evaluated at 30, 60, 90, and 120 min after treatment.

#### 4.3.12. Radiographic Analysis

A digital Faxitron machine (Ultrafocus; Faxitron Bioptics, Tucson, AZ) was used to image the femurs of all mice (all groups in chronic studies) on days 0 (baseline), 7, 10, and 14 post-surgery to determine an average bone score and to exclude the data for any mice with full cortical fracture before day 14. Two blinded observers evaluated the images using a 5-point bone rating scale. Mice were anesthetized with 80 mg/kg ketamine to 12 mg/kg xylazine (in a 10 mL/kg volume) on day 0 before the surgery, D7, D10, and D14, and radiographs were acquired. The bone scoring scale was as follows: 0 = normal bone, 1 = 1–3 lesions with no fracture, 2 = 4+ lesions with no fracture, 3 = uni-cortical, full thickness fracture, and 4 = bicortical, full thickness fracture. Animals with bicortical, full fractures were euthanized as our IACUC protocol measures dictated.

### 4.4. Statistical Analysis

Statistical significance for all data was analyzed using one-way ANOVA with Tukey’s post hoc correction or repeated measure two-way ANOVA with Bonferroni post hoc correction, depending on the experimental design. Data were reported as mean ± SEM for n = 10–12 mice /treatment group. Power analyses were performed on cumulated data using G* Power 3.1 software to estimate the required number; we found adequate statistical separation for each group to detect 80% between groups at alpha ˂ 0.05. Data were expressed with data analysis performed using GraphPad Prism 7.0 (Graph pad INC., San Diego, CA, USA). All experiments were performed with treatment-blinded observers’ drug vs. vehicle treatment to increase rigor and data collection responsibility.

## Figures and Tables

**Figure 1 ijms-24-15007-f001:**
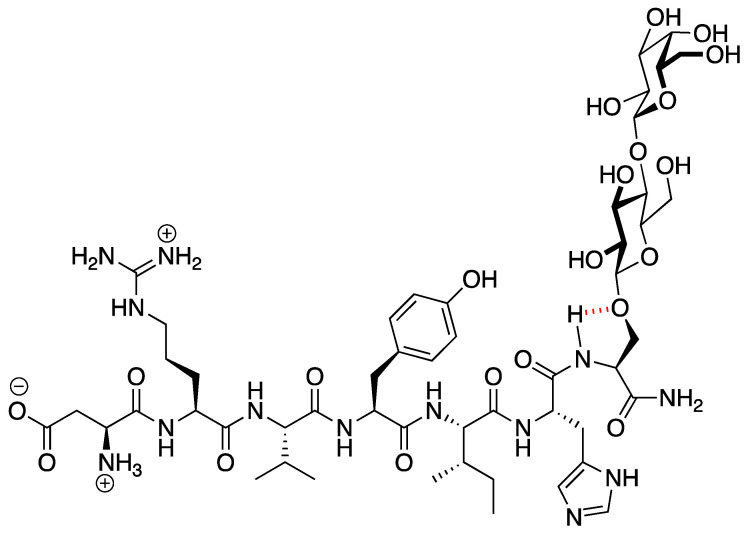
The chemical structure of PNA6.

**Figure 2 ijms-24-15007-f002:**
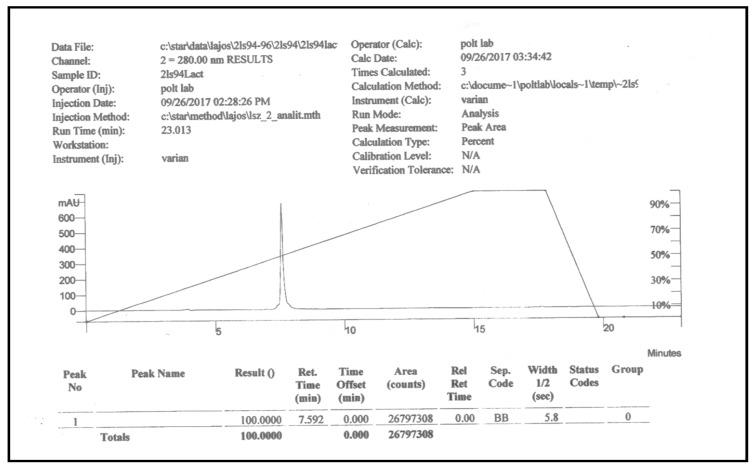
Analytical HPLC chromatogram of pure PNA6.

**Figure 3 ijms-24-15007-f003:**
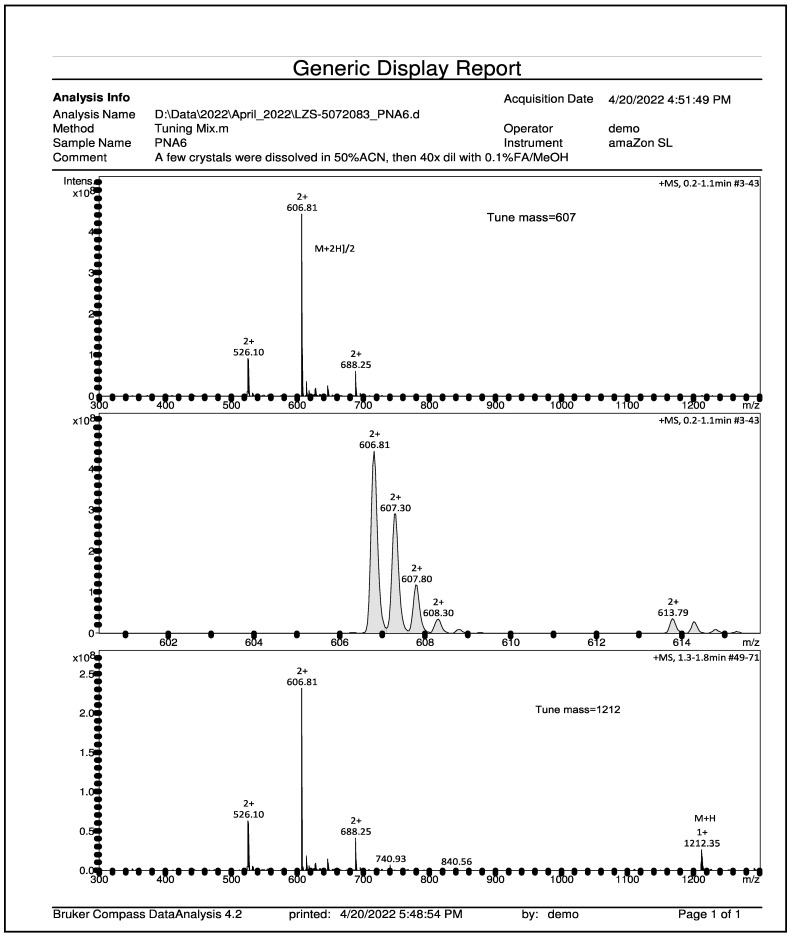
Mass spectroscopy spectrum of pure PNA6 shows single charged ion ([M + H]^+^ 1212.35) and double charged ion ([M + 2H]^2+^ 606.81).

**Figure 4 ijms-24-15007-f004:**
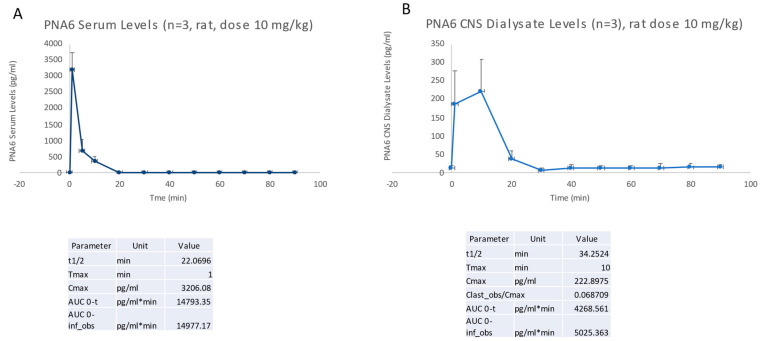
PK analysis of PNA6 in serum (**A**) and brain parenchyma (**B**). Data are presented as mean ± standard error (SEM).

**Figure 5 ijms-24-15007-f005:**
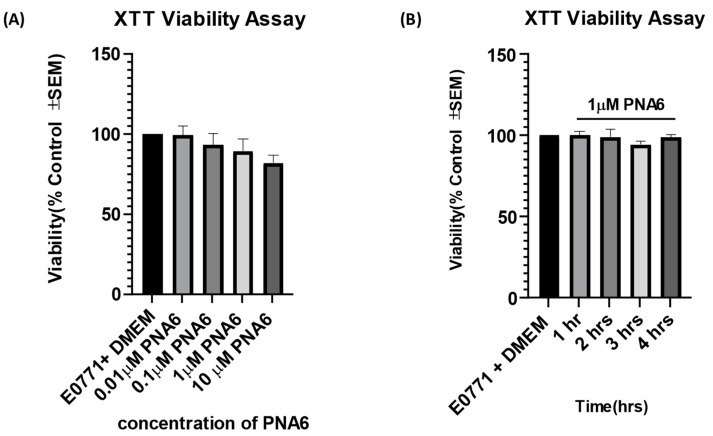
Effects of PNA6 on E0771 cancer cell viability in vitro. (**A**) E0771 adenocarcinoma cells do not have a significant change in viability when treated with varying concentrations of PNA6 for 24 h; the cell viability was tested using the XTT assay. One-Way ANOVA with Bonferroni post hoc correction (n = 5). (**B**) Timeline of 1, 2, 3, and 4 h post PNAS application at 1µM utilizing the XTT assay. There was no significant effect on the cell viability of E0771 cells in the presence of PNA6 from 1 to 4 h.

**Figure 6 ijms-24-15007-f006:**
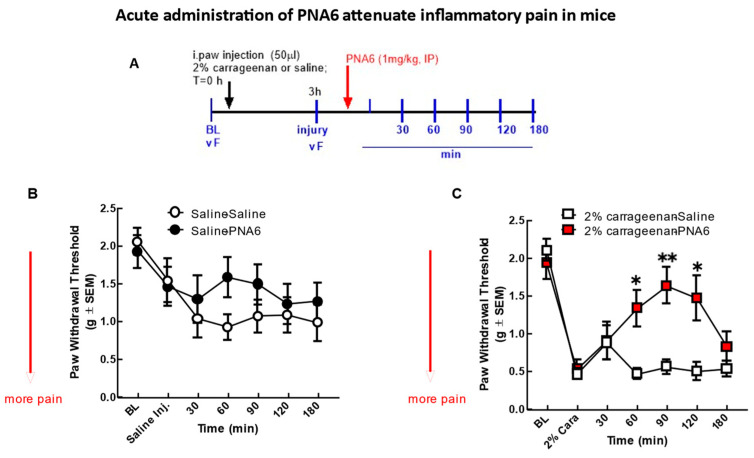
Acute model of inflammatory pain. Significant mechanical allodynia was detected after 2% λ-carrageenan injection. (**A**) Timeline of experiment. (**B**) Mice injected intra-hind paw with saline control did not show a significant decrease in mechanical thresholds in the ipsilateral hind paw. Neither PNA6 nor saline given intraperitoneally 3 h posts i-paw injection resulted in a significant change over the 3 h testing period. (**C**) 2% λ-carrageenan injection significantly decreased hind-paw thresholds 3 h post-injection. PNA6 (1 mg/kg i.p.) significantly reversed the mechanical hypersensitivity induced by 2% λ-carrageenan in the ipsilateral hind paw at 60, 90, and 120 min (n = 10–12 mice in each group, * *p* < 0.05, ** *p* < 0.01). Saline (1 mL/kg, i.p.) did not significantly affect 2% λ-carrageenan-induced mechanical hypersensitivity. Results were analyzed by repeated measures (RM) two-way ANOVA followed by the Bonferroni post hoc test.

**Figure 7 ijms-24-15007-f007:**
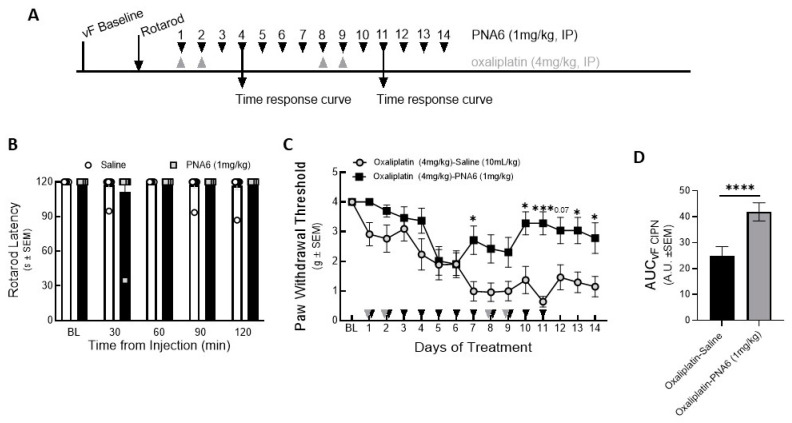
Chronic effect of PNA6 on CIPN. Timeline of experiment (**A**). PNA6 does not impair animal locomotion at all testing times (**B**). Animals experienced a significant decrease in mechanical thresholds due to the treatment of oxaliplatin (4 mg/kg i.p on days 1, 2, 8, and 9). Animals with oxaliplatin demonstrated a significant increase in the paw withdrawal thresholds on days 7, 10, 11, 13, and 14 after PNA6 (1 mg/kg i.p.) treatment for 14 consecutive days (**C**). The area under the curve was calculated, demonstrating significant inhibition of CIPN by PNA6 (**D**). Values represent mean ± SEM, n = 12 per group, * *p* < 0.05, *** *p* < 0.001, **** *p* < 0 0.0001. Results were analyzed by repeated measures (RM) two-way ANOVA followed by Bonferroni’s multiple comparison test.

**Figure 8 ijms-24-15007-f008:**
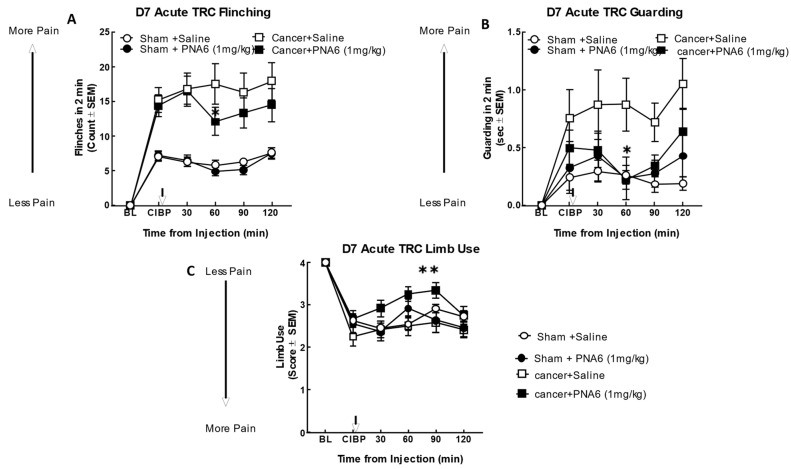
Acute administration of PNA6 in established cancer-induced bone pain attenuates spontaneous pain behavior in CIBP. Animals display no pain behavior before surgery (BL = baseline). Instead, animals demonstrate increased flinching, guarding, and decreased limb use post-cancer surgery inoculation (CIBP). (**A**) Time-response curve (TRC) of flinching behavior after PNA6 1 mg/kg at times 30, 60, 90, and 120 min on day seven after surgery. (**B**) Guarding behavior time-response curve after 1 mg/kg at times 30, 60, 90, and 120 min on day seven after surgery. (**C**) Time-response curve of limb use at 30, 60, 90, and 120 min on day seven after surgery. Saline had no significant effect on all three behaviors. The time of peak effect for PNA6 was 60 min after administration. n = 10–12 mice in each group, * *p* < 0.05, ** *p* < 0.01. Results were analyzed using RM, two-way ANOVA followed by Bonferroni post hoc test.

**Figure 9 ijms-24-15007-f009:**
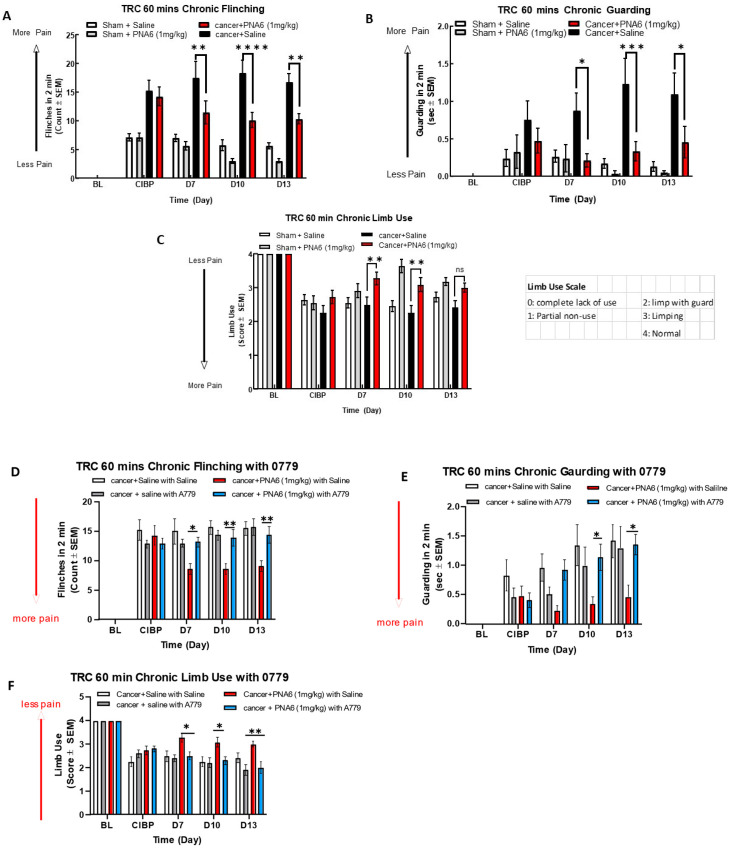
Sustained administration of PNA6 attenuates cancer pain in the murine model of cancer-induced bone pain. Spontaneous pain behavior (**A**) flinching, (**B**) guarding, and (**C**) limb use measured before and at regular intervals following cancer inoculation in female C57BL/6J mice. At D7 post-surgery, animals inoculated with E0771 breast adenocarcinoma cells had significantly elevated spontaneous pain behavior in mice in comparison to the sham mice group. PNA6 at D7, 10, and 13 significantly attenuated flinching and guarding in mice with cancer compared to saline-treated mice (**A**,**B**). PNA6 at D7 and 10 significantly increased limb use in mice with cancer compared to saline-treated mice (**C**). MasR1 antagonist, A779, reversed the antinociception of PNA6 in the murine model of cancer-induced bone pain. To investigate receptor dependence, animals were dosed post-surgery from days 7 to 13 with A779 1 mg/ kg 30 min before administration of PNA6. Spontaneous pain behaviors flinching (**D**), guarding (**E**), and limb use (**F**) were recorded as previously described. Daily administration of A779 reversed the effect of PNA6 in the CIBP model. In all behavioral tests, n = 10–12 mice in each group, * *p* < 0.05, ** *p* < 0.01, *** *p* < 0.001, **** *p* < 0.0001, ns meant no significant difference. Results were analyzed using RM two-way ANOVA followed by Bonferroni post hoc test. Limb Uue scale: 0 = complete lack of use, 1 = partial non-use, 2 = limp with guard, 3 = limping, 4 = normal.

**Figure 10 ijms-24-15007-f010:**
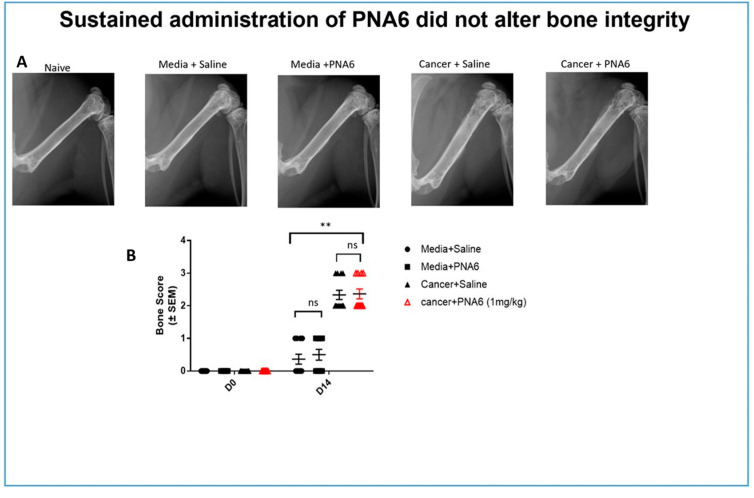
Sustained administration of PNA6 did not alter bone integrity. (**A**) Representative radiograph of the right femur of mice D14 post-surgery. Radiographs were obtained before surgery and at D14 post-surgery to monitor cancer-induced lesions. (**B**) Quantification of bone scores (n = 10–12/treatment groups). Bone scores were determined on a 5-point scale by blinded observers. The scoring system was as follows: 0 = normal bone, 1 = 1–3 lesions with no fracture, 2 = 4–6 lesions with no fracture, 3 = uni-cortical, full thickness fracture, and 4 = bi-cortical, full thickness fracture. Data are expressed as mean ± S.E.M. ** *p* < 0.01, ns meant no significant difference (one-way ANOVA, Tukey’s HSD post hoc).

## Data Availability

The original contributions presented in the study are included in the article/supplementary material; further inquiries can be directed to the corresponding author.

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
