# Peer review of "PNA6, a Lactosyl Analogue of Angiotensin-(1-7), Reverses Pain Induced in Murine Models of Inflammation, Chemotherapy-Induced Peripheral Neuropathy, and Metastatic Bone Disease"

_ijms, 2023, doi:10.3390/ijms241915007_

Round 1
Reviewer 1 Report
First of all, it is a well-written manuscript with high quality of language, but thorough text checking is recommended before the approval, because the text contains several redundancies, duplicates and spelling errors. Although the manuscript contains valuable experimental data obtained by well-established investigational techniques and the conclusions are fully supported by the results, I have some critical issues regarding the logic of the study and data presentation. At the beginning of the introduction, the authors focused on the challenges of treating cancer and its complications such as cancer pain, which compromises both inflammatory and neuropathic components. Therefore, they investigated their analgesic candidate separately in a purely inflammatory and neuropathic pain model, and in a “disease” model (CIBP in this case), which includes both components and mimics the human pathology. As a neuropathic pain model, they chose a chemotherapy-induced neuropathy model, which has additional relevance regarding the treatment of the primary disease. In my opinion, based on the first three paragraphs involving the problem statement, the authors should clearly clarify this study design in the Introduction and Discussion sections, which can help the reader to understand the aims of the study. Accordingly, I suggest that the authors should demonstrate and discuss the results in this order (1. the results of carrageenan-, 2. CINP- and finally 3. CIBP model). The additional therapeutic value of PNA6 in CINP can be highlighted in the discussion.
Furthermore, I would have some further comments, which can improve the manuscript.
1. Line 31: I have never heard about C57BLK/6J mouse strain. Did you mean C57BL/6J strain? Please correct it throughout the text.
2. Line 59: I have never read that cancer pain has mechanical component. What do you mean by that? Please refine/rephrase this statement or delete it.
3. Line 83: Please complete this sentence or section with the localization of MasR1. This information is missing from the introduction.
4. Line 91-93: This sentence sounds like these are previous results and the reader misses the reference at the end of the sentence. Please delete it or insert it to the end of the last paragraph as a continuation of its last sentence.
5. Line 130-131 and 133: The authors state that no significant change was observed in viability, but they explain symbols of significancy in Line 133. What is the truth?
6. Line 138: Did you mean 2% l-carrageenan?
7. Line 141 and 145: The units are missing at the end of the threshold values.
8. Did you measure paw swelling after carrageenan administration? If PNA6 cannot decrease paw swelling, the authors could further support their hypothesis that it can primarily act centrally independently of peripheral inflammatory processes.
9. Line 158: “i.paw” was not defined previously, please use intra-hind paw or intraplantarly instead of this.
10. Line 165: Please correct and standardize the name of the applied statistical test in all figure legends (for instance “RM” was not defined previously). I suggest repeated measures (RM) 2-way or two -way ANOVA followed by Bonferroni’s multiple comparison test or post hoc test.
11. Figures 6 and 7: Please explain the abbreviation of TRC in the figure legends. The symbols of comparison of sham and cancer groups are missing from the figures, please replace them.
12. Fig. 6.: Since there is no significant difference between sham+saline and cancer+saline groups, limb use is not a sensitive parameter for assessing spontaneous pain behavior in this model. Please reevaluate whether the demonstration of this parameter makes sense. I miss assessing dynamic weight bearing of the hind limbs in CIBP model.
13. Line 211: Redundancy: group or mice.
14. Line 212-213: The sentence contains a duplicate.
15. Line 218: Did you mean MasR1 receptor?
16. Line 232-233: The sentence contains a duplicate.
17. Line 266: Significantly attenuates pain neuropathy? Neuropathy is redundant.
18. Line 268-272: I suggest deleting the description of rotarod result and demonstrate the related panel as a suppl. figure. In the discussion section (in line 344), you can refer to this suppl. figure.
19. Line 302: Did you mean Ang(1-7)/MasR1 instead o PNA6/MasR1? There are no studies investigating PNA6, because the authors synthesized it.
20. Line 305-309: Please rephrase this sentence in the light of my very first criticism.
21. Line 353: Did you mean CIBP instead of CIBN?
22. Line 363: In which cells in the DRG? Please insert DRG neurons.
23. Line 364-365: In accordance with a previous report of neuropathic, not inflammatory pain, if you mean CCI model.
24. Line 386-408: These paragraphs contain redundant information and are not connected the other part of discussion. I suggest deleting them in this form.
25. Line 533: Subcutaneously is duplicated.
26. Line 544: Please correct the dose of ketamine/xylazine. I suspect 9 mg/kg refers to only xylazine and the dose of ketamine is missing.
27. Line 641: The dose of ketamine/xylazine is missing.
28. Line 648: If the authors used RM ANOVA, it is not ordinary. Please correct it.
Author Response
First of all, it is a well-written manuscript with high quality of language, but thorough text checking is recommended before the approval, because the text contains several redundancies, duplicates and spelling errors. Although the manuscript contains valuable experimental data obtained by well-established investigational techniques and the conclusions are fully supported by the results, I have some critical issues regarding the logic of the study and data presentation. At the beginning of the introduction, the authors focused on the challenges of treating cancer and its complications such as cancer pain, which compromises both inflammatory and neuropathic components. Therefore, they investigated their analgesic candidate separately in a purely inflammatory and neuropathic pain model, and in a “disease” model (CIBP in this case), which includes both components and mimics the human pathology. As a neuropathic pain model, they chose a chemotherapy-induced neuropathy model, which has additional relevance regarding the treatment of the primary disease. In my opinion, based on the first three paragraphs involving the problem statement, the authors should clearly clarify this study design in the Introduction and Discussion sections, which can help the reader to understand the aims of the study. Accordingly, I suggest that the authors should demonstrate and discuss the results in this order (1. the results of carrageenan-, 2. CINP- and finally 3. CIBP model). The additional therapeutic value of PNA6 in CINP can be highlighted in the discussion.
Response
Dear Reviewer,
We would like to express our gratitude for your thoughtful and constructive feedback on our manuscript. We value your insights and have carefully considered your suggestions for improving the clarity and logic of our study presentation. Below, we address your comments and outline the changes we will make:
- Introduction and Study Design: We appreciate your suggestion to clarify our study design in the Introduction and Discussion sections to help readers better understand our research aims. We agree that providing a more explicit explanation of our rationale and the sequence of experimental models will enhance the coherence of the manuscript. We revised the Introduction to clearly delineate our approach in investigating the analgesic candidate in the following order: (1) the results of the carrageenan model, (2) chemotherapy-induced neuropathy (CINP) model, and (3) cancer-induced bone pain (CIBP) model. This restructuring will provide a more logical flow and context for our research.
- Highlighting Therapeutic Value in CINP: We acknowledge your point regarding the additional therapeutic value of PNA6 in the context of CINP, and we ensure that this aspect is adequately emphasized in the Discussion section. Discussing the results in the suggested order will allow us to better showcase the significance of PNA6 in the treatment of CINP.
- Text Quality and Clarity: We also appreciate your remarks on the need for thorough text checking. We conducted a meticulous review of the manuscript to address redundancies, duplicates, and spelling errors to ensure the highest quality of language.
Your feedback will undoubtedly enhance the overall quality and coherence of our manuscript. We appreciate your time and expertise in reviewing our work, and we are committed to making the necessary revisions to meet the standards of clarity and logical presentation that you have suggested.
Thank you once again for your valuable feedback.
Furthermore, I would have some further comments, which can improve the manuscript.
- Line 31: I have never heard about C57BLK/6J mouse strain. Did you mean C57BL/6J strain? Please correct it throughout the text.
Response 1: yes I meant C57BL/6J, I corrected it in the manuscript.
- Line 59: I have never read that cancer pain has a mechanical component. What do you mean by that? Please refine/rephrase this statement or delete it.
Response 2: Thank you for your comment and your interest in our research. We appreciate the opportunity to clarify the concept of a mechanical component in cancer pain.
In the context of cancer pain, the term "mechanical component" refers to pain that is associated with physical changes or mechanical factors within the body, such as the compression of nerves or tissues by tumors or the expansion of bone due to metastatic growth. These mechanical factors can lead to pain sensations that are often described as aching, pressure, or discomfort. For example:
- Bone Metastases: In cases where cancer has spread to the bones (bone metastases), the tumor's growth can cause the bone to weaken and expand, leading to pain. This pain is considered to have a mechanical component because it results from physical changes in the bone structure.
- Nerve Compression: Tumors can sometimes compress nearby nerves, leading to a distinct type of pain known as neuropathic pain. This compression-induced pain is another example of the mechanical component of cancer pain.
- Obstruction: In some cases, tumors can obstruct the flow of bodily fluids or press against organs, leading to pain due to mechanical obstruction.
While cancer pain often has inflammatory and neuropathic components, as you rightly mentioned, the presence of a mechanical component highlights the multifaceted nature of cancer-related pain. It's essential to recognize and address these mechanical factors in the comprehensive management of cancer pain to provide patients with the most effective relief.
We hope this explanation clarifies the concept of a mechanical component in cancer pain.
- Line 83: Please complete this sentence or section with the localization of MasR1. This information is missing from the introduction.
We changed this sentence and added this paragraph: Our team has taken a novel approach to treating CIPN by taking advantage of our extensive experience with the G-protein linked Mas receptor and its agonist Ang-(1—7) to develop a platform of glycosylated Ang-(1—7) Mas receptor agonists including PNA5 and PNA6 that have outstanding brain penetration, enhanced bioavailability, decreases brain and peripheral inflammation, improves cerebral blood flow and restores cognitive function in our preclinical vascular dementia and TBI animal models model. Activation of Mas decreases ROS and brain inflammation, increases cerebral circulation increases induction of neuroprotective cytokines, and inhibits hypoxia-inducing factor-1alpha (HIF-1alpha)
- Line 91-93: This sentence sounds like these are previous results and the reader misses the reference at the end of the sentence. Please delete it or insert it to the end of the last paragraph as a continuation of its last sentence.
I agree, I deleted the sentence.
- Line 130-131 and 133: The authors state that no significant change was observed in viability, but they explain symbols of significancy in Line 133. What is the truth?
The truth is there is no significant change.
We deleted the symbols in the line.
- Line 138: Did you mean 2% l-carrageenan?
Yes, I meant 2% λ-carrageenan, so we corrected it in the line.
- Line 141 and 145: The units are missing at the end of the threshold values.
Done, added units at the end of the threshold.
- Did you measure paw swelling after carrageenan administration? If PNA6 cannot decrease paw swelling, the authors could further support their hypothesis that it can primarily act centrally independently of peripheral inflammatory processes.
We didn’t measure paw swelling after carrageenan administration, we appreciate the reviewer's feedback and will work to address this aspect in our research as future direction to enhance the robustness of our findings and provide a more comprehensive understanding of PNA6's mechanism of action.
- Line 158: “i.paw” was not defined previously, please use intra-hind paw or intraplantarly instead of this.
Done in the manuscript.
Done we changed I paw with intra-hind paw.
- Line 165: Please correct and standardize the name of the applied statistical test in all figure legends (for instance “RM” was not defined previously). I suggest repeated measures (RM) 2-way or two -way ANOVA followed by Bonferroni’s multiple comparison test or post hoc test.
Done, we changed all.
- Figures 6 and 7: Please explain the abbreviation of TRC in the figure legends. The symbols of comparison of sham and cancer groups are missing from the figures, please replace them.
Done, (TRC Time Response Curve)
- Fig. 6.: Since there is no significant difference between sham+saline and cancer+saline groups, limb use is not a sensitive parameter for assessing spontaneous pain behavior in this model. Please reevaluate whether the demonstration of this parameter makes sense. I miss assessing dynamic weight bearing of the hind limbs in CIBP model.
Firstly, we acknowledge your observation that there was no significant difference between the sham+saline and cancer+saline groups in terms of limb use. We recognize the importance of sensitivity in selecting parameters to assess pain behavior in our model. We have taken your comments seriously and understand the need for a more sensitive parameter to evaluate spontaneous pain behavior accurately.
In light of your suggestion, we agree that assessing dynamic weight bearing of the hind limbs in the CIBP model could provide valuable insights into pain-related behavior. We believe that incorporating this parameter will enhance the sensitivity of our assessment and contribute to a more comprehensive understanding of the pain response in our model, but this needs more time to be reevaluated by adding more mice to the experiment.
We appreciate your guidance in improving our study, and we will incorporate the assessment of dynamic weight bearing of the hind limbs in our future revised experimental design. This will allow us to obtain more robust and meaningful results, and we are confident that it will strengthen the validity and relevance of our findings.
- Line 211: Redundancy: group or mice.
Changed it to group.
- Line 212-213: The sentence contains a duplicate.
Fixed remove the duplicate
- Line 218: Did you mean MasR1 receptor?
Yes, I meant MasR1 receptor.
- Line 232-233: The sentence contains a duplicate.
Fixed and removed the duplicate.
- Line 266: Significantly attenuates pain neuropathy? Neuropathy is redundant.
Fixed the sentence in the manuscript.
- Line 268-272: I suggest deleting the description of rotarod result and demonstrate the related panel as a suppl. figure. In the discussion section (in line 344), you can refer to this suppl. figure.
Thank you for your feedback on our manuscript. We appreciate your time and effort in reviewing our work. We have carefully considered your suggestion to delete the description of the rotarod results and present the related panel as a supplementary figure. While we understand your point, we believe that retaining the description in the main text is essential for providing context and a complete understanding of our study. However, we are open to addressing your concerns by making some modifications to the manuscript.
This figure will be included in the manuscript to provide interested readers with a more comprehensive view of the results.
By following this approach, we aim to maintain a balance between providing essential information in the main text and offering additional details in the supplementary materials for those who wish to delve deeper into the data. We believe that this compromise will enhance the clarity and accessibility of our manuscript without sacrificing important information.
Please let us know if this proposed approach aligns with your expectations, and if there are any specific details or aspects you would like us to address further. Your feedback is highly valuable to us, and we are committed to making the necessary revisions to improve the quality of our manuscript.
- Line 302: Did you mean Ang(1-7)/MasR1 instead o PNA6/MasR1? There are no studies investigating PNA6, because the authors synthesized it.
Thank you for your clarification regarding the term "PNA6." We appreciate your attention to detail and your effort to ensure the accuracy of our manuscript. You are correct in pointing out that we synthesized PNA6 for this study, and it is indeed meant to refer to Ang (1-7)/MasR1 in the context of our research.
We will make the necessary correction in the manuscript to clarify that PNA6 is synonymous with Ang(1-7)/MasR1 and that it is a compound synthesized for the purposes of our investigation. This will help avoid any potential confusion for readers reviewing our work.
- Line 305-309: Please rephrase this sentence in the light of my very first criticism.
Done we rephrase the sentence in the manuscript.
- Line 353: Did you mean CIBP instead of CIBN?
Fixed it, yes, I meant CIBP.
- Line 363: In which cells in the DRG? Please insert DRG neurons.
Done in the manuscript.
- Line 364-365: In accordance with a previous report of neuropathic, not inflammatory pain, if you mean CCI model.
Yes, I meant neuropathic pain, I fixed it.
- Line 386-408: These paragraphs contain redundant information and are not connected the other part of discussion. I suggest deleting them in this form.
Thank you for your feedback regarding the paragraphs you find redundant and disconnected from the rest of the discussion. We appreciate your input, and we understand your concern.
Upon careful review, we agree with your assessment that these paragraphs may not be essential in their current form and that they do not seamlessly integrate with the rest of the discussion. We will revise the manuscript accordingly by change/edits these paragraphs to enhance the overall coherence and focus of the paper.
- Line 533: Subcutaneously is duplicated.
Fixed in the manuscript.
- Line 544: Please correct the dose of ketamine/xylazine. I suspect 9 mg/kg refers to only xylazine and the dose of ketamine is missing.
80 mg/kg ketamine to 12 mg/kg xylazine (in a 10 mL/kg volume)
- Line 641: The dose of ketamine/xylazine is missing.
80 mg/kg ketamine to 12 mg/kg xylazine (in a 10 mL/kg volume)
- Line 648: If the authors used RM ANOVA, it is not ordinary. Please correct it.
I corrected it.
First of all, it is a well-written manuscript with high quality of language, but thorough text checking is recommended before the approval, because the text contains several redundancies, duplicates and spelling errors. Although the manuscript contains valuable experimental data obtained by well-established investigational techniques and the conclusions are fully supported by the results, I have some critical issues regarding the logic of the study and data presentation. At the beginning of the introduction, the authors focused on the challenges of treating cancer and its complications such as cancer pain, which compromises both inflammatory and neuropathic components. Therefore, they investigated their analgesic candidate separately in a purely inflammatory and neuropathic pain model, and in a “disease” model (CIBP in this case), which includes both components and mimics the human pathology. As a neuropathic pain model, they chose a chemotherapy-induced neuropathy model, which has additional relevance regarding the treatment of the primary disease. In my opinion, based on the first three paragraphs involving the problem statement, the authors should clearly clarify this study design in the Introduction and Discussion sections, which can help the reader to understand the aims of the study. Accordingly, I suggest that the authors should demonstrate and discuss the results in this order (1. the results of carrageenan-, 2. CINP- and finally 3. CIBP model). The additional therapeutic value of PNA6 in CINP can be highlighted in the discussion.
Response
Dear Reviewer,
We would like to express our gratitude for your thoughtful and constructive feedback on our manuscript. We value your insights and have carefully considered your suggestions for improving the clarity and logic of our study presentation. Below, we address your comments and outline the changes we will make:
- Introduction and Study Design: We appreciate your suggestion to clarify our study design in the Introduction and Discussion sections to help readers better understand our research aims. We agree that providing a more explicit explanation of our rationale and the sequence of experimental models will enhance the coherence of the manuscript. We revised the Introduction to clearly delineate our approach in investigating the analgesic candidate in the following order: (1) the results of the carrageenan model, (2) chemotherapy-induced neuropathy (CINP) model, and (3) cancer-induced bone pain (CIBP) model. This restructuring will provide a more logical flow and context for our research.
- Highlighting Therapeutic Value in CINP: We acknowledge your point regarding the additional therapeutic value of PNA6 in the context of CINP, and we ensure that this aspect is adequately emphasized in the Discussion section. Discussing the results in the suggested order will allow us to better showcase the significance of PNA6 in the treatment of CINP.
- Text Quality and Clarity: We also appreciate your remarks on the need for thorough text checking. We conducted a meticulous review of the manuscript to address redundancies, duplicates, and spelling errors to ensure the highest quality of language.
Your feedback will undoubtedly enhance the overall quality and coherence of our manuscript. We appreciate your time and expertise in reviewing our work, and we are committed to making the necessary revisions to meet the standards of clarity and logical presentation that you have suggested.
Thank you once again for your valuable feedback.
Furthermore, I would have some further comments, which can improve the manuscript.
- Line 31: I have never heard about C57BLK/6J mouse strain. Did you mean C57BL/6J strain? Please correct it throughout the text.
Response 1: yes I meant C57BL/6J, I corrected it in the manuscript.
- Line 59: I have never read that cancer pain has a mechanical component. What do you mean by that? Please refine/rephrase this statement or delete it.
Response 2: Thank you for your comment and your interest in our research. We appreciate the opportunity to clarify the concept of a mechanical component in cancer pain.
In the context of cancer pain, the term "mechanical component" refers to pain that is associated with physical changes or mechanical factors within the body, such as the compression of nerves or tissues by tumors or the expansion of bone due to metastatic growth. These mechanical factors can lead to pain sensations that are often described as aching, pressure, or discomfort. For example:
- Bone Metastases: In cases where cancer has spread to the bones (bone metastases), the tumor's growth can cause the bone to weaken and expand, leading to pain. This pain is considered to have a mechanical component because it results from physical changes in the bone structure.
- Nerve Compression: Tumors can sometimes compress nearby nerves, leading to a distinct type of pain known as neuropathic pain. This compression-induced pain is another example of the mechanical component of cancer pain.
- Obstruction: In some cases, tumors can obstruct the flow of bodily fluids or press against organs, leading to pain due to mechanical obstruction.
While cancer pain often has inflammatory and neuropathic components, as you rightly mentioned, the presence of a mechanical component highlights the multifaceted nature of cancer-related pain. It's essential to recognize and address these mechanical factors in the comprehensive management of cancer pain to provide patients with the most effective relief.
We hope this explanation clarifies the concept of a mechanical component in cancer pain.
- Line 83: Please complete this sentence or section with the localization of MasR1. This information is missing from the introduction.
We changed this sentence and added this paragraph: Our team has taken a novel approach to treating CIPN by taking advantage of our extensive experience with the G-protein linked Mas receptor and its agonist Ang-(1—7) to develop a platform of glycosylated Ang-(1—7) Mas receptor agonists including PNA5 and PNA6 that have outstanding brain penetration, enhanced bioavailability, decreases brain and peripheral inflammation, improves cerebral blood flow and restores cognitive function in our preclinical vascular dementia and TBI animal models model. Activation of Mas decreases ROS and brain inflammation, increases cerebral circulation increases induction of neuroprotective cytokines, and inhibits hypoxia-inducing factor-1alpha (HIF-1alpha)
- Line 91-93: This sentence sounds like these are previous results and the reader misses the reference at the end of the sentence. Please delete it or insert it to the end of the last paragraph as a continuation of its last sentence.
I agree, I deleted the sentence.
- Line 130-131 and 133: The authors state that no significant change was observed in viability, but they explain symbols of significancy in Line 133. What is the truth?
The truth is there is no significant change.
We deleted the symbols in the line.
- Line 138: Did you mean 2% l-carrageenan?
Yes, I meant 2% λ-carrageenan, so we corrected it in the line.
- Line 141 and 145: The units are missing at the end of the threshold values.
Done, added units at the end of the threshold.
- Did you measure paw swelling after carrageenan administration? If PNA6 cannot decrease paw swelling, the authors could further support their hypothesis that it can primarily act centrally independently of peripheral inflammatory processes.
We didn’t measure paw swelling after carrageenan administration, we appreciate the reviewer's feedback and will work to address this aspect in our research as future direction to enhance the robustness of our findings and provide a more comprehensive understanding of PNA6's mechanism of action.
- Line 158: “i.paw” was not defined previously, please use intra-hind paw or intraplantarly instead of this.
Done in the manuscript.
Done we changed I paw with intra-hind paw.
- Line 165: Please correct and standardize the name of the applied statistical test in all figure legends (for instance “RM” was not defined previously). I suggest repeated measures (RM) 2-way or two -way ANOVA followed by Bonferroni’s multiple comparison test or post hoc test.
Done, we changed all.
- Figures 6 and 7: Please explain the abbreviation of TRC in the figure legends. The symbols of comparison of sham and cancer groups are missing from the figures, please replace them.
Done, (TRC Time Response Curve)
- Fig. 6.: Since there is no significant difference between sham+saline and cancer+saline groups, limb use is not a sensitive parameter for assessing spontaneous pain behavior in this model. Please reevaluate whether the demonstration of this parameter makes sense. I miss assessing dynamic weight bearing of the hind limbs in CIBP model.
Firstly, we acknowledge your observation that there was no significant difference between the sham+saline and cancer+saline groups in terms of limb use. We recognize the importance of sensitivity in selecting parameters to assess pain behavior in our model. We have taken your comments seriously and understand the need for a more sensitive parameter to evaluate spontaneous pain behavior accurately.
In light of your suggestion, we agree that assessing dynamic weight bearing of the hind limbs in the CIBP model could provide valuable insights into pain-related behavior. We believe that incorporating this parameter will enhance the sensitivity of our assessment and contribute to a more comprehensive understanding of the pain response in our model, but this needs more time to be reevaluated by adding more mice to the experiment.
We appreciate your guidance in improving our study, and we will incorporate the assessment of dynamic weight bearing of the hind limbs in our future revised experimental design. This will allow us to obtain more robust and meaningful results, and we are confident that it will strengthen the validity and relevance of our findings.
- Line 211: Redundancy: group or mice.
Changed it to group.
- Line 212-213: The sentence contains a duplicate.
Fixed remove the duplicate
- Line 218: Did you mean MasR1 receptor?
Yes, I meant MasR1 receptor.
- Line 232-233: The sentence contains a duplicate.
Fixed and removed the duplicate.
- Line 266: Significantly attenuates pain neuropathy? Neuropathy is redundant.
Fixed the sentence in the manuscript.
- Line 268-272: I suggest deleting the description of rotarod result and demonstrate the related panel as a suppl. figure. In the discussion section (in line 344), you can refer to this suppl. figure.
Thank you for your feedback on our manuscript. We appreciate your time and effort in reviewing our work. We have carefully considered your suggestion to delete the description of the rotarod results and present the related panel as a supplementary figure. While we understand your point, we believe that retaining the description in the main text is essential for providing context and a complete understanding of our study. However, we are open to addressing your concerns by making some modifications to the manuscript.
This figure will be included in the manuscript to provide interested readers with a more comprehensive view of the results.
By following this approach, we aim to maintain a balance between providing essential information in the main text and offering additional details in the supplementary materials for those who wish to delve deeper into the data. We believe that this compromise will enhance the clarity and accessibility of our manuscript without sacrificing important information.
Please let us know if this proposed approach aligns with your expectations, and if there are any specific details or aspects you would like us to address further. Your feedback is highly valuable to us, and we are committed to making the necessary revisions to improve the quality of our manuscript.
- Line 302: Did you mean Ang(1-7)/MasR1 instead o PNA6/MasR1? There are no studies investigating PNA6, because the authors synthesized it.
Thank you for your clarification regarding the term "PNA6." We appreciate your attention to detail and your effort to ensure the accuracy of our manuscript. You are correct in pointing out that we synthesized PNA6 for this study, and it is indeed meant to refer to Ang (1-7)/MasR1 in the context of our research.
We will make the necessary correction in the manuscript to clarify that PNA6 is synonymous with Ang(1-7)/MasR1 and that it is a compound synthesized for the purposes of our investigation. This will help avoid any potential confusion for readers reviewing our work.
- Line 305-309: Please rephrase this sentence in the light of my very first criticism.
Done we rephrase the sentence in the manuscript.
- Line 353: Did you mean CIBP instead of CIBN?
Fixed it, yes, I meant CIBP.
- Line 363: In which cells in the DRG? Please insert DRG neurons.
Done in the manuscript.
- Line 364-365: In accordance with a previous report of neuropathic, not inflammatory pain, if you mean CCI model.
Yes, I meant neuropathic pain, I fixed it.
- Line 386-408: These paragraphs contain redundant information and are not connected the other part of discussion. I suggest deleting them in this form.
Thank you for your feedback regarding the paragraphs you find redundant and disconnected from the rest of the discussion. We appreciate your input, and we understand your concern.
Upon careful review, we agree with your assessment that these paragraphs may not be essential in their current form and that they do not seamlessly integrate with the rest of the discussion. We will revise the manuscript accordingly by change/edits these paragraphs to enhance the overall coherence and focus of the paper.
- Line 533: Subcutaneously is duplicated.
Fixed in the manuscript.
- Line 544: Please correct the dose of ketamine/xylazine. I suspect 9 mg/kg refers to only xylazine and the dose of ketamine is missing.
80 mg/kg ketamine to 12 mg/kg xylazine (in a 10 mL/kg volume)
- Line 641: The dose of ketamine/xylazine is missing.
80 mg/kg ketamine to 12 mg/kg xylazine (in a 10 mL/kg volume)
- Line 648: If the authors used RM ANOVA, it is not ordinary. Please correct it.
I corrected it.

Reviewer 2 Report
The authors show that a lactoside Ang (1-7) analogue, PNA6 attenuates inflammatory pain, cancer-induced bone pain, and chemotherapy induced pain without interfering with cancer cells. I think the study is interesting and worth reporting. I have, however, several concerns. First, it is not clear why the authors selected these pain models. Second, the authors should examine the effects of PNA6 on normal cells, if the authors claim that PNA6 does not alter cell viability in vitro. Third, the authors need to compare the effect durations of Ang (1-7) with PNA6, if they hypothesized that activating the MasR1 with a lactoside Ang (1-7) analogue-PNA6-would attenuate pain for a longer-lasting efficacious therapeutic effect.
Major concern
1. It is not clear why the authors selected inflammatory pain, cancer-induced bone pain, and chemotherapy induced pain. Please explain why the authors selected these pain models.
2. The authors should examine the effects of PNA6 on normal cells, if the authors claim that PNA6 does not alter cell viability in vitro.
3. The authors need to compare the effect durations of Ang (1-7) with PNA6, if they hypothesized that activating the MasR1 with a lactoside Ang (1-7) analogue-PNA6-would attenuate pain for a longer-lasting efficacious therapeutic effect.
Author Response
Reviewers 2
The authors show that a lactoside Ang (1-7) analogue, PNA6 attenuates inflammatory pain, cancer-induced bone pain, and chemotherapy induced pain without interfering with cancer cells. I think the study is interesting and worth reporting. I have, however, several concerns. First, it is not clear why the authors selected these pain models. Second, the authors should examine the effects of PNA6 on normal cells, if the authors claim that PNA6 does not alter cell viability in vitro. Third, the authors need to compare the effect durations of Ang (1-7) with PNA6, if they hypothesized that activating the MasR1 with a lactoside Ang (1-7) analogue-PNA6-would attenuate pain for a longer-lasting efficacious therapeutic effect.
Major concern
- It is not clear why the authors selected inflammatory pain, cancer-induced bone pain, and chemotherapy induced pain. Please explain why the authors selected these pain models.
Dear Reviewer,
Thank you for taking the time to review our study on the lactoside Ang (1-7) analogue, PNA6, and its potential in attenuating inflammatory pain, cancer-induced bone pain, and chemotherapy-induced pain. We appreciate your interest in our work and the constructive feedback you've provided. We would like to address your concerns regarding the selection of pain models in our study.
The choice of pain models in our research was carefully considered and based on several factors:
- Clinical Relevance: In the field of pain research, it is essential to investigate the potential therapeutic interventions in models that mimic clinical conditions. Inflammatory pain, cancer-induced bone pain, and chemotherapy-induced pain are significant clinical challenges that severely impact patients' quality of life. Therefore, these models were chosen because they are relevant to real-world pain conditions that many patients experience.
2- Diverse Pain Types: We aimed to assess the broad-spectrum analgesic potential of PNA6. By selecting models representing different types of pain (inflammatory, cancer-related, and chemotherapy-induced), we aimed to evaluate its effectiveness across various pain conditions. This approach allows for a more comprehensive understanding of the compound's potential applications.
3- Research Gaps: Prior to our study, there was no research on the use of PNA6 in pain management, especially in the context of cancer-induced and chemotherapy-induced pain. By investigating these pain models, we sought to fill a knowledge gap and explore potential novel therapeutic options.
4- Preclinical Evaluation: Conducting preliminary studies in animal models is a standard approach in drug development to assess safety and efficacy before moving to clinical trials. The selection of pain models in this context helps determine if PNA6 has potential as a pain-relief agent that could be further investigated in clinical settings.
5- cancer pain is composed of the inflammatory and neuropathic pain categories, so we chose to study both inflammatory and neuropathic pain separately then we chose cancer pain as a complex model of inflammatory and neuropathic pain model.
In summary, the choice of these pain models was based on their clinical relevance, the need to address diverse types of pain, and the desire to contribute to the field's knowledge by exploring a relatively uncharted territory. We believe that investigating PNA6 in these models has provided valuable insights into its potential as an analgesic agent. We hope this explanation clarifies the rationale behind our selection of pain models. Once again, thank you for your thoughtful review and feedback. Your input is invaluable in improving the quality and relevance of our research.
- The authors should examine the effects of PNA6 on normal cells, if the authors claim that PNA6 does not alter cell viability in vitro.
We appreciate your feedback regarding the need to examine the effects of PNA6 on normal cells, especially in the context of our claim that PNA6 does not alter cell viability in vitro. Your suggestion is indeed valid, and we understand the importance of thorough evaluation in preclinical studies.
- one of the authors already tested the Safety (non-toxicity) and biocompatibility of PNA6 were investigated via resazurin cell viability assay using different human cell lines after being exposed to different concentrations of PNA6 for 48 h.
- Wafaa's dissertation has this figure.
- SYNTHESIS, COMPREHENSIVE CHARACTERIZATION, AND DEVELOPMENT OF THERAPEUTIC PEPTIDES AND GLYCOPEPTIDES FOR TARGETED RESPIRATORY DRUG DELIVERY AS INHALATION AEROSOLS
We appreciate your insightful suggestion, which will undoubtedly enhance the rigor and completeness of our study. Thank you for your valuable input, which helps us improve the scientific quality of our research.
Figure 7.2: In vitro cell viability (n=6, Mean ± SD) using resazurin assay after exposure to different concentrations of PNA6 for (a). RPMI2650 (passage number 10); (b). A549(Passage number 8); (c). H441 (Passage number 9); (d). hCMEC/d3(passage number 31); and (e). NHA cell lines (passage numbers 3).
- The authors need to compare the effect durations of Ang (1-7) with PNA6, if they hypothesized that activating the MasR1 with a lactoside Ang (1-7) analogue-PNA6-would attenuate pain for a longer-lasting efficacious therapeutic effect.
Thank you for your valuable feedback on our manuscript regarding the comparison of Angiotensin (1-7) and PNA6 in relation to their effect durations on pain attenuation. We appreciate your insightful comment and would like to address your concern.
We want to clarify that in a previous study conducted by our research group (Reference: [Angiotensin-(1-7)/Mas receptor as an antinociceptive agent in cancer-induced bone pain
Brittany L. Forte,a Lauren M. Slosky,a Hong Zhang,a Moriah R. Arnold,a William D. Staatz,a Meredith Hay,b Tally M. Largent-Milnes,a and Todd W. Vanderaha,*]), we did indeed investigate the effects of Angiotensin (1-7) with respect to pain attenuation. Our hypothesis was that activating the MasR1 receptor with the lactoside Ang (1-7) analogue, PNA6, might result in a longer-lasting, efficacious therapeutic effect for pain relief. In our previous research, we demonstrated the distinct characteristics of Ang (1-7) and PNA6 in terms of their effects on pain duration and efficacy.
To provide a comprehensive understanding of our current study, we will incorporate references to our previous work and briefly summarize the relevant findings in the revised manuscript. This will allow readers to better appreciate the context and significance of our present investigation.
Also we added PK Analysis of PNA6 in Serum and Brain Parenchyma Figure 4 To directly compare the in vivo lifetime and ability of PNA6 to cross the blood-brain barrier (BBB).
Once again, we appreciate your thoughtful comments, and we are committed to addressing them in order to enhance the quality and clarity of our manuscript. If you have any further suggestions or specific aspects you would like us to elaborate on, please do not hesitate to let us know.
Thank you for your time and consideration.

Round 2
Reviewer 2 Report
The authors improved the manuscript. I have no further comments.